# LiveMCPBench: Can Agents Navigate an Ocean of MCP Tools?

## Abstract

Model Context Protocol (MCP) has become a key infrastructure for connecting LLMs with external tools, scaling to 10,000+ MCP servers with diverse tools. Unfortunately, there is still a large gap between real-world MCP usage and current evaluation: they typically assume single-server settings and directly inject tools into the model's context, bypassing the challenges of large-scale retrieval and multi-tool composition. To bridge this gap, we propose **LiveMCPBench**, which evaluates 95 real-world daily tasks explicitly constructed to stress diverse tools and scaled multi-server routing. The benchmark includes a ready-to-deploy tool suite of 70 servers with 527 tools, ensuring reproducibility without scattered API configuration. We further introduce an LLM-as-a-Judge evaluation framework that directly verifies task outcomes, handling dynamic data sources and multiple valid solution paths. We benchmark 10 state-of-the-art LLMs and observe a substantial performance gap: while Claude-Sonnet-4 reaches 78.95% task success, most models achieve only 30–50%. Our analysis reveals that active tool composition strongly correlates with task success, whereas retrieval errors account for nearly half of all failures—highlighting retrieval as the dominant bottleneck. Together, these results provide the first large-scale, reproducible diagnosis of MCP agent capabilities and point towards future research on improving retrieval robustness and encouraging effective tool composition. Code and data will be released upon publication.

## 1 Introduction

Tool-use agents powered by large language models (LLMs) are emerging as a critical step toward general intelligence (Qu et al., 2025; Wang et al., 2024). The Model Context Protocol (Anthropic, 2024, MCP) has rapidly become the core infrastructure for connecting LLMs with external tools (Ehtesham et al., 2025). Its ecosystem is rapidly scaling to more than 10,000 servers with diverse functionalities (Hou et al., 2025). Meanwhile, pretrained models are being equipped with the ability to invoke MCP servers directly (Qwen, 2025b). MCP has greatly expanded the boundaries of tool-use agents, enabling them to tackle increasingly complex real-world tasks (Ray, 2025). As a result, the ability to use MCP effectively has become a key criterion of an agent's capability.

Despite the rapid expansion of MCP, existing benchmarks fall short of capturing its real-world challenges, as shown in Figure 1. Traditional tool-use benchmarks such as API-Bank (Li et al., 2023) and ToolBench (Qin et al., 2024) rely on simulated or unstable APIs, resulting in less realistic and unreliable tasks. For example, 55.6% of APIs in ToolBench are no longer available (Guo et al., 2024). In contrast, MCP provides a more stable invocation interface through a package-like management model, where servers can be reliably maintained and provide both local and online functionality. MCP adopts a server-tool architecture, where each server exposes multiple related tools with contextual descriptions, providing richer semantics. This structure requires agents to develop retrieval and composition abilities that go beyond conventional tool search. However, current MCP benchmarks remain narrow in scope. For example, MCPBench (Luo et al., 2025) covers only 10 servers and directly injects tools into the LLM's context, bypassing large-scale retrieval and multi-tool composition entirely. Furthermore, most existing MCP benchmarks are still conducted in static environments, making them unable to evaluate agents' performance in dynamic, large-scale MCP ecosystems.

These limitations lead to two central research questions:

- *RQ1: How can we systematically evaluate an agent's retrieval and composition capabilities in large-scale, diverse MCP ecosystems?*

- *RQ2: How can we design evaluation methodologies that are both scalable and reproducible under dynamic, real-world settings?*

To address these questions, we introduce **LiveMCPBench**, a benchmark for realistic, large-scale MCP evaluation. LiveMCPBench covers six practical domains and consists of 95 multi-step, daily tasks that explicitly test retrieval and composition abilities. We focus on daily scenarios because most MCP servers are designed to provide up-to-date information (e.g., news), making them a natural and challenging frontier for assessing MCP-use capabilities. To facilitate reproducibility and ease of use, we provide **LiveMCPTool**, a curated collection of 70 MCP servers with 527 tools, packaged as a ready-to-use environment that removes the need for configuring numerous scattered API keys. For evaluation, we propose **LiveMCPEval**, an automatic evaluation framework based on the LLM-as-a-Judge (Zheng et al., 2023), enabling scalable assessment even under dynamic data sources (e.g., evolving weather or news) and supporting multiple valid solution trajectories. Crucially, it supports scalable and open-ended evaluation, allowing seamless expansion as new MCP servers and tasks are introduced, without costly manual annotation.

Building on LiveMCPBench, we develop **MCP Copilot Agent**, a ReACT agent (Yao et al., 2023) capable of retrieving and composing tools across a large MCP toolset. We evaluate 10 leading LLMs on the benchmark and find a substantial performance gap: while Claude-Sonnet-4 achieves a 78.95% task success rate, the majority of widely used frontier models remain in the 30—50% range. By analyzing agent behaviors, we observe that active tool composition strongly correlates with task success, indicating that models capable of flexibly combining tools are more reliable. A detailed error analysis further shows that nearly half of all failures stem from tool retrieval errors, establishing retrieval as the primary bottleneck in large-scale MCP usage. These findings highlight the urgent need for future systems that jointly **improve tool composition** and **mitigate retrieval errors**. To further validate our evaluation pipeline, we conduct human annotation of

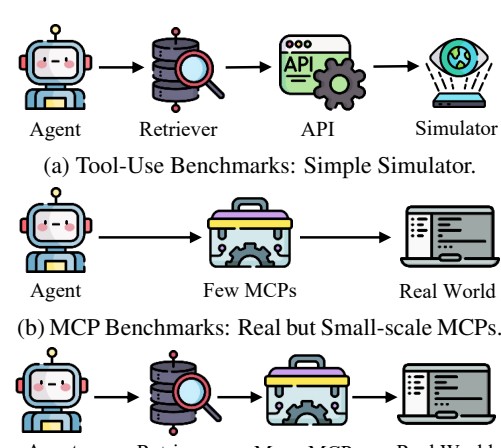

(a) Tool-Use Benchmarks: Simple Simulator.

(b) MCP Benchmarks: Real but Small-scale MCPs.

**(c) LiveMCPBench: Real & Large-scale MCPs.**

Figure 1: LiveMCPBench focuses on daily task resolution using a large-scale MCP toolset.

agent trajectories and assess human–model agreement. DeepSeek-V3 attains an agreement rate of 81.05%, demonstrating the reliability of our evaluation protocol.

In summary, our key contributions are:

- We introduce LiveMCPBench, a benchmark with 95 multi-step daily tasks that test agents' large-scale retrieval and multi-tool composition abilities in large-scale MCP ecosystems.

- We provide a reproducible toolset (70 MCP servers, 527 tools) together with an LLM-as-a-Judge automatic evaluation framework, enabling scalable, automated, and human-aligned assessment in dynamic environments.

- Our empirical study offers a large-scale diagnosis of MCP agent capabilities, highlighting tool retrieval and composition as unsolved challenges and laying the groundwork for future research.

Figure 2: LiveMCPBench comprises four parts: (1) Diverse Daily Task, a collection of 95 daily tasks; (2) LiveMCPTool, a large-scale MCP toolset; (3) MCP Copilot Agent, a ReACT-based agent supporting retrieval and multi-turn tool invocation across the MCP toolset; (4) LiveMCPEval, an automated LLM-as-a-Judge evaluation system for accurately assessing dynamic tasks.

# 2 LIVEMCPBENCH

We present LiveMCPBench (Figure 2), a benchmark designed to evaluate the ability of agent systems to retrieve appropriate tools from a large-scale MCP toolset for accomplishing general-purpose daily tasks. Building such a benchmark requires addressing three key challenges:

- **Designing representative tasks** that faithfully capture and expose the unique characteristics and inherent complexities of large-scale MCP ecosystems.

- **Constructing a comprehensive, reproducible toolset** that remains both large and functionally complete while being easy to deploy for consistent and fair benchmarking.

- **Automating robust evaluation** of agent performance on dynamic, evolving online tasks, where task context and available tools may change over time.

In this section, we first detail our task construction methodology. Then, we describe the collection process of the LiveMCPTool. Next, we introduce LiveMCPEval, an LLM-as-a-Judge evaluation framework that enables robust and scalable assessment. Finally, we build MCP Copilot Agent, which serves as our baseline approach.

## 2.1 TASK CONSTRUCTION

To ensure that LiveMCPBench reflects the practical strengths of MCP servers, we design tasks grounded in everyday scenarios where these servers are most impactful. Since many MCP servers are oriented toward real-time information retrieval (e.g., news, stock prices, weather updates) or local office productivity (e.g., spreadsheet analysis, scheduling), daily-life settings naturally highlight their distinctive capabilities. By anchoring tasks in such contexts, we capture both the immediacy and utility that characterize MCP-driven interactions.

The resulting benchmark spans six daily domains: *Office* (e.g., spreadsheet analysis), *Lifestyle* (e.g., news retrieval), *Leisure* (e.g., video game inquiries), *Finance* (e.g., stock price monitoring), *Travel* (e.g., ticket search), and *Shopping* (e.g., product recommendations). Tasks in these domains are inherently dynamic, often depending on evolving contexts such as monitoring fluctuating markets. They are also long-horizon, as solving them typically requires coordinating multiple tools rather than a single call. At the same time, the tasks retain genuine utility, reflecting realistic and meaningful user needs that go beyond artificial benchmarks.

Table 1: Comparison with existing MCP and tool-use benchmarks. **Plug & Play** indicates whether the toolset can be used directly without additional API keys. **Dynamic** denotes whether the task depending on evolving contexts. **Syn.** denotes tasks generated by LLMs. Note that **MCP-Zero** is not a benchmark, but contains a toolset. And, it is not directly usable, as many of its tools require API keys.

| | Tools | | | | Tasks | | | |
|---|---|---|---|---|---|---|---|---|
| | Stable | Servers | Tools | Plug & Play | Type | Number | Dynamic | Evaluation |
| *Tool-Use Benchmarks* | | | | | | | | |
| API-Bank (Li et al., 2023) | ✓ | 8 | 73 | ✓ | Real | 314 | ✗ | Rule |
| ToolBench (Qin et al., 2024) | ✗ | 49 | 3,451 | ✓ | Syn. | 126,486 | ✓ | LLM |
| $\tau$-bench (Yao et al., 2025) | ✓ | 2 | 28 | ✓ | Real | 165 | ✗ | Rule |
| *MCP Benchmarks* | | | | | | | | |
| MCPBench (Luo et al., 2025) | ✓ | 10 | 10 | ✗ | Real | 911 | ✗ | Rule |
| MCP-RADAR (Gao et al., 2025) | ✓ | 9 | 42 | ✓ | Real | 300 | ✗ | Rule |
| MCP-Zero (Fei et al., 2025) | ✓ | 308 | 2,797 | ✗ | - | - | - | - |
| MCPEval (Liu et al., 2025) | ✓ | 12 | 77 | ✗ | Syn. | 676 | ✓ | LLM |
| **LiveMCPBench (*ours*)** | ✓ | 70 | 527 | ✓ | Real | 95 | ✓ | LLM |

The task creation process employed a rigorous two-stage methodology involving two groups of computer science students serving as task *proposers* and *validators*. Each group consisted of three members to ensure both diversity of perspectives and reliability of annotation. *Proposers* first generated scenario-specific tasks based on personal experience, with LLM-assisted ideation permitted but strictly vetted for authenticity. Each proposer then interacted with our toolset to complete their proposed task, meticulously annotating key points to preserve the task's compositional depth. *Validators* subsequently scrutinized both the task design and corresponding toolchain invocations, eliminating duplicates while enforcing quality standards. In total, *proposers* initially produced 300 candidate tasks across six domains. After iterative validation and refinement, this pipeline yielded 95 high-fidelity daily tasks (see Appendix E for annotation principles and distribution statistics).

## 2.2 LIVEMCPTOOL COLLECTION

While prior study (Hou et al., 2025) suggests the existence of over 10,000 MCP servers, creating a practical and accessible toolset remains nontrivial due to critical usability constraints. The predominant challenge stems from dependency fragmentation: the majority of MCP servers necessitate proprietary API keys or integrations with third-party services, rendering them impractical for a standardized toolset. To address this, we introduce a rigorously validated methodology for constructing a high-quality, dependency-free MCP toolset—prioritizing reproducibility and broad applicability. Our approach first aggregates 5,588 server configurations from a MCP Marketplace[1], then systematically filters out key-dependent servers to eliminate access barriers.

Beyond accessibility, we ensure the toolset's representativeness through structured curation and expert annotation. Tools are taxonomically organized into five functional categories (Discovery, Visualization, File Access, Location, and Miscellaneous), followed by manual vetting to exclude low-quality implementations. This two-stage pipeline yields 70 MCP servers providing 527 tools, each verified for standalone functionality and categorical relevance. By decoupling the toolset from external dependencies, our collection establishes a reproducible toolset for large-scale MCP performance analysis (see Appendix F for distribution details).

## 2.3 LIVEMCPEVAL

Evaluating agent trajectories in dynamic environments is non-trivial, as task outcomes often depend on evolving contexts, and multiple tool invocation trajectories may lead to equally valid solutions. Such properties make traditional metrics, such as exact tool-matching accuracy, inadequate. To address this, we introduce LiveMCPEval, an automatic evaluation framework that leverages an

---

[1]`mcp.so`

LLM-as-a-Judge system. Instead of relying on fixed ground-truth trajectories, LiveMCPEval directly verifies task completion by considering tool usage patterns and contextual feedback. A central component of this design is the use of *key points*—critical subtasks or intermediate conditions that must be satisfied for success—which can be either manually annotated or automatically extracted by LLMs (see Appendix H for examples). In our benchmark, all tasks are equipped with a verified set of key points to ensure reliable evaluation. Formally, given a task $T$, a set of key points $P$, an execution trajectory $A$ (including retrievals and tool calls), and tool descriptions $D$, the evaluator performs binary classification to determine task success:

$$\mathcal{J} = Evaluator(T, P, A, D) \tag{1}$$

where $\mathcal{J} \in \{\text{Success}, \text{Failure}\}$. The overall benchmark metric is the *success rate* across tasks. Beyond accuracy, the framework is designed for openness and scalability: as new MCP servers and tasks emerge, LiveMCPEval enables seamless benchmark expansion without costly manual annotations, ensuring its long-term relevance in MCP ecosystems. A systematic comparison with existing benchmarks is presented in Table 1.

## 2.4 MCP COPILOT AGENT

Daily tasks are inherently dynamic, making fixed pipelines for tool use impractical. For instance, always retrieving before executing may fail when outputs must be combined across tools or when re-routing is required after errors. Agents must instead adapt to evolving contexts.

We formulate tool retrieval and invocation as a Partially Observable Markov Decision Process (Silver & Veness, 2010, POMDP), where decisions rely only on tool descriptions and execution feedback. The environment is defined by: (1) hidden states $\mathcal{S}$ for the underlying task; (2) observations $\mathcal{O}$ from retrieved descriptions and feedback; (3) a language action space $\mathcal{A}$; (4) transition $\mathcal{T} : \mathcal{S}_t \times \mathcal{A} \rightarrow \mathcal{S}_{t+1}$; (5) reward $\mathcal{R} : \mathcal{S} \rightarrow \mathbb{R}$ for task success. The action space $\mathcal{A}$ consists of three operations: *Route*, which retrieves $k$ candidate tools (fixed at $k = 5$ in our experiments) from the full toolset; *Execute*, which invokes a selected tool with specified parameters and returns its output as feedback; *Response*, which terminates the process and delivers the final result to the user (see Appendix G for more detail). Tool score in *Route* follows MCP-Zero (Fei et al., 2025):

$$\text{score} = (s_{\text{server}} \times s_{\text{tool}}) \times \max(s_{\text{server}}, s_{\text{tool}}) \tag{2}$$

where $s_{\text{server}}$ and $s_{\text{tool}}$ denote cosine similarity to server-level and tool-level descriptions. This form emphasizes joint alignment while letting server priors dominate when tool matches are weak.

Our agent builds on ReACT, integrating reasoning and action in a unified loop. Unlike conventional APIs treating tools as isolated black boxes, MCP introduces a server–tool hierarchy that demands structural reasoning and coordination. The MCP Copilot Agent thus adapts flexibly under uncertainty while exploiting MCP's hierarchical richness.

## 3 EXPERIMENTS AND RESULTS

### 3.1 SETUP

We evaluate 10 frontier models: Claude-Opus-4 and Claude-Sonnet-4 (Anthropic, 2025), GPT-4.1 and GPT-4.1-Mini (Openai, 2025), Gemini-2.5-Pro (Google, 2025), Deepseek-V3 and Deepseek-R1 (DeepSeek-AI, 2025), Qwen3-235B-A22B and Qwen3-32B (Qwen, 2025a), and Qwen2.5-72B-Instruct (Qwen et al., 2025). For evaluation, we employ Deepseek-V3. We use the Qwen3-Embedding-0.6B model (Zhang et al., 2025) to obtain vector representations of both the queries and the MCPs (further implementation details are provided in Appendix G).

### 3.2 MAIN RESULTS

Table 2 reports the task success rates of ten frontier models. Our key findings are as follows:

1. **Meta-Tool-Learning in Claude Models.** The Claude series exhibits striking meta-tool-learning proficiency, with Claude-Sonnet-4 and Claude-Opus-4 achieving success rates of 78.95% and 70.53% respectively. These results demonstrate their strong ability to flexibly composition tools from a large-scale toolset to accomplish complex real-world tasks.

Table 2: Task success rate results for the frontier models. Evaluation using Deepseek-V3 with human-annotated key points.

| Model | Office | Leisure | Travel | Lifestyle | Finance | Shopping | Overall (%) |
|---|---|---|---|---|---|---|---|
| Claude-Sonnet-4-20250514 | 90.32 | 64.29 | 75.00 | 80.00 | 78.57 | 66.67 | 78.95 |
| Claude-Opus-4-20250514 | 80.65 | 64.29 | 66.67 | 86.67 | 64.29 | 33.33 | 70.53 |
| DeepSeek-R1-0528 | 41.94 | 50.00 | 58.33 | 46.67 | 50.00 | 55.56 | 48.42 |
| Qwen3-235B-A22B | 54.84 | 35.71 | 41.67 | 53.33 | 50.00 | 44.44 | 48.42 |
| GPT-4.1-Mini | 45.16 | 50.00 | 50.00 | 46.67 | 42.86 | 22.22 | 44.21 |
| Qwen2.5-72B-Instruct | 35.48 | 35.71 | 50.00 | 40.00 | 57.14 | 55.56 | 43.16 |
| DeepSeek-V3-0324 | 41.94 | 42.86 | 50.00 | 40.00 | 28.57 | 55.56 | 42.11 |
| Gemini-2.5-Pro | 48.39 | 28.57 | 16.67 | 60.00 | 57.14 | 11.11 | 41.05 |
| GPT-4.1 | 51.61 | 28.57 | 25.00 | 46.67 | 35.71 | 22.22 | 38.95 |
| Qwen3-32B | 29.03 | 14.29 | 25.00 | 53.33 | 28.57 | 33.33 | 30.53 |

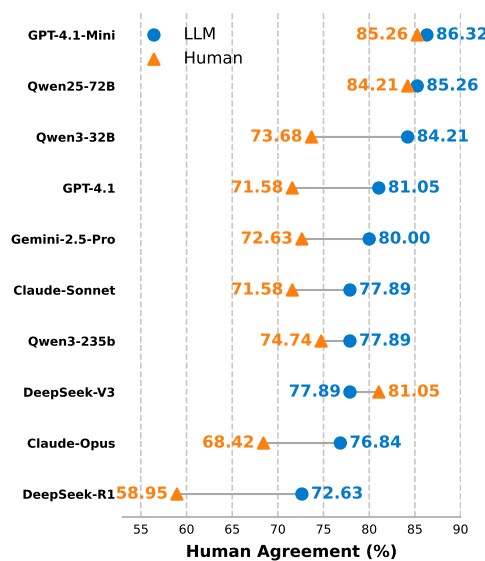

Figure 3: Correlation between model evaluation performance and human agreement across evaluators, based on annotated agent trajectories from Claude-Opus-4 and Claude-Sonnet-4.

Figure 4: Comparison of human agreement rates across different models when evaluated using human-annotated versus LLM-generated key points.

2. **Performance Variance Across Models.** We observe substantial variance in performance: while most contemporary models achieve only 30%–50% task success rates, the Claude series substantially outperforms them. This performance gap underscores fundamental limitations in the meta-tool-learning capabilities of other models.

3. **Domain-Specific Superiority of Claude Models.** The Claude series exhibits particularly dominant performance in *Office* and *Lifestyle* scenarios, surpassing competing models by more than 30%. This domain-specific advantage highlights Claude's unique adaptability and practical utility in real-world MCP environments.

**Is LiveMCPEval Reliable?** We evaluated the reliability of LiveMCPEval by comparing its automatic judgments with human annotations on execution trajectories of top models (Claude-Sonnet-4, Claude-Opus-4) and by testing baseline models as evaluators (Figure 3). Results show that LiveMCPEval is accurate under appropriate evaluators: Deepseek-V3 reaches 78.95% human agreement, while GPT-4.1 Mini and Qwen2.5-72B-Instruct achieve ∼75%. In contrast, Deepseek-R1, Claude-Opus-4, and Qwen3-32B yield lower agreement (60–70%), likely due to difficulty handling long trajectories. Overall, LiveMCPEval is dependable, but its effectiveness depends on the model.

Table 3: Performance efficiency metrics: Steps denotes average dialogue turns, Tools indicates average used tools, *execute* represents average tool executions, *route* refers to average retrievals, Tokens refers to total tokens consumed (in millions), and Overall shows average task success rate.

| Model | Steps | Tools | *excute* | *route* | Tokens (M) | Overall (%) |
|---|---|---|---|---|---|---|
| Claude-Sonnet-4 | 20.09 | 2.71 | 5.59 | 2.98 | 6.07 | 78.95 |
| Claude-Opus-4 | 25.53 | 3.40 | 6.93 | 4.35 | 7.16 | 70.53 |
| Qwen3-235B | 16.76 | 1.59 | 5.12 | 1.77 | 3.73 | 48.42 |
| DeepSeek-R1 | 10.33 | 1.24 | 2.11 | 2.00 | 2.88 | 48.42 |
| GPT-4.1-Mini | 10.89 | 1.37 | 2.71 | 1.65 | 2.96 | 44.21 |
| Qwen2.5-72B | 11.22 | 1.31 | 2.80 | 1.38 | 3.55 | 43.16 |
| DeepSeek-V3 | 8.33 | 1.01 | 1.29 | 1.41 | 2.11 | 42.11 |
| Gemini-2.5-Pro | 8.08 | 0.99 | 1.46 | 1.35 | 2.03 | 41.05 |
| GPT-4.1 | 9.03 | 1.31 | 1.72 | 1.64 | 2.48 | 38.95 |
| Qwen3-32B | 9.99 | 1.16 | 2.31 | 1.19 | 1.93 | 30.53 |

**Does LiveMCPEval Generalize?** We tested the generalizability of LiveMCPEval on Claude-Sonnet-4's trajectories by examining how LLM-generated key points affect human agreement (Figure 4, examples in Appendix H). Results show that LiveMCPEval generalizes well across models: even without human-annotated references, most evaluators achieved higher agreement using automatically generated key points. This demonstrates the framework's openness to new tasks and tools, since auto-generated key points can adaptively serve as substitutes when human references are unavailable. Moreover, Deepseek-V3 best leverages human-annotated key points, suggesting advantages for scenarios where such references are available.

## 4 ANALYSIS

### 4.1 EFFICIENCY ANALYSIS

To compare the behavioral characteristics of different models, we present the average number of dialogue turns, used tools, tool execution attempts, and retrieval calls in Table 3. Based on these metrics, we draw the following conclusions:

1. **Active tool composition correlates with stronger performance.** Claude-Sonnet-4, with 2.71 tools, 5.59 executions, and 2.98 retrievals, achieves the highest success rate (78.95%), far outperforming conservative models such as Gemini-2.5-Pro or DeepSeek-V3, which use only a single tool on average and remain below 42%. This suggests that proactive interaction with the tool environment is a key mechanism for effective MCP usage.

2. **Efficiency hinges on balancing exploration and exploitation.** While Claude-Opus-4 explores even more aggressively (3.40 tools, 6.93 executions), its success rate drops to 70.53%, indicating diminishing returns and potential error accumulation. Conversely, models that under-explore (e.g., Qwen3-32B at 1.16 tools and 30.53% success) fail to adapt. Thus, performance is not determined by raw tool counts but by how effectively a model manages the exploration-exploitation trade-off.

In practical applications, a trade-off between model performance and cost must be carefully considered. To provide actionable insights for model selection, we plotted the relationship between logarithmic cost and performance, along with the corresponding Pareto frontier (Lotov et al., 2004). As illustrated in Figure 5, our analysis reveals two key findings:

1. **Near-Linear Trade-off on the Pareto Frontier.** The performance and logarithmic cost of models along the Pareto frontier exhibit an approximately linear relationship. This observation presents a valuable opportunity for optimizing cost-performance balance in real-world tool-calling agents.

2. **Optimal Cost-Performance Models.** The models positioned on the Pareto frontier represent the most cost-effective choices for tool calling. These include Qwen3-32B, Qwen2.5-

72B-Instruct, Deepseek-R1-0528, and Claude-Sonnet-4, each demonstrating distinct advantages in terms of cost-performance efficiency.

## 4.2 ERROR ANALYSIS

We conducted a detailed error analysis on the trajectories of current retrieval and invocation agents to provide insights for future development. Human annotators were employed to classify error types in the trajectories of Claude-Opus-4 and Claude-Sonnet-4. Based on the modules in the MCP Copilot Agent framework, we identified four distinct and easily distinguishable error categories (Figure 6). Each erroneous trajectory was uniquely classified into one error type without overlap. Detailed error examples are provided in Appendix I. Our error analysis centers on the agent trajectories, complementary evaluation analysis is presented in Appendix H.

**Query Error.** Query errors occur when the generated query either lacks semantic relevance to the required tools or exhibits a granularity mismatch with tool capabilities. For instance, in the task "summarize today's news and save as PDF," the agent might request a single omnipotent tool despite the availability of specialized tools for news retrieval and PDF generation. Such granularity mismatches prevent the retrieval system from providing appropriate tools, and agents often fail to refine queries based on retrieval feedback. Hallucinated queries for irrelevant tools further exacerbate this issue. These errors stem from limitations in LLMs' task decomposition and planning capabilities, suggesting room for improvement despite their generally competent performance.

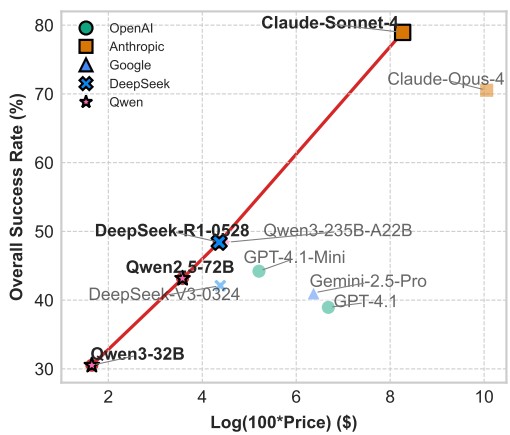

Figure 5: Log-Price vs. Performance scatter plot with Pareto frontier representation. Different colors represent different model families.

**Retrieve Error.** Retrieve errors arise when semantically appropriate queries fail to match available tools due to retrieval system shortcomings. For example, in the task "Convert the YouTube video to MP3 format," the retrieval system may overlook the *youtube downloader* tool (which supports format conversion) due to unrecognized semantic equivalence between "convert to MP3" and the tool's documented "extract audio tracks" functionality. These errors highlight challenges in hierarchical retrieval (e.g., MCP server-tool structures) and semantic similarity computation. Dominating the error distribution, retrieval errors underscore the critical need for enhanced retrieval architectures and more robust similarity metrics.

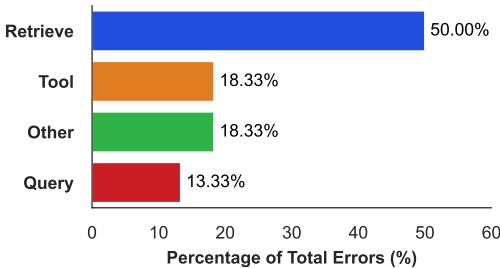

Figure 6: Distribution of errors for four types. We counted trajectories of Claude series models.

**Tool Error.** Tool errors occur when the agent retrieves the correct tool but invokes it incorrectly— e.g., via error parameters or incomplete server/tool names. In the task "summarize news and save to specified path," the agent might supply "path name" instead of the required "path" parameter to the save tool. Such inaccuracies reflect limitations in contextual precision and memory retention. While modern LLMs exhibit strong contextual understanding, these errors indicate a need for more sophisticated memory mechanisms to ensure reliable tool usage.

**Other Error.** This category encompasses sporadic failures beyond the above types, including network timeouts or model invocation errors. For example, in "summarize today's news," a network timeout during news retrieval may cause the agent to abandon the task without retries or alterna-

tive solutions. Such behavior reveals deficiencies in framework design, particularly the absence of robust error-handling mechanisms (e.g., failure recovery, adaptive tool exploration). The prevalence of these errors suggests that while current frameworks support basic exploration, significant improvements in fault tolerance and proactive problem-solving are needed.

## 5 RELATED WORK

### 5.1 TOOL-USE BENCHMARKS

Most existing benchmarks for tool use rely on simulated APIs, given the instability of real-world interfaces. For example, API-Bank (Li et al., 2023) and $\tau$-bench (Yao et al., 2025) construct artificial toolsets to ensure stability, while ToolAlpaca (Tang et al., 2023) and Seal-Tools (Wu et al., 2025) collect real-world APIs but cannot execute actual calls. A third line of work, such as ToolBench (Qin et al., 2024) and ShortcutsBench (SHEN et al., 2025), attempts to integrate real APIs, but frequent interface changes often render tools unusable (Guo et al., 2024).

More recent efforts, e.g., StableToolBench-MirrorAPI (Guo et al., 2025), use fine-tuned LLMs to simulate APIs and calls. Yet, these benchmarks remain largely API-centric, inheriting instability and lacking support for broader functionality—such as direct manipulation of local files or interactions with local software.

The emergence of MCP offers a new paradigm: a stable, unified interface for building general-purpose toolsets. In this work, we leverage MCP to construct a practical toolset that overcomes both API instability and functional limitations, enabling a comprehensive and reliable real-world tool-use benchmark.

### 5.2 MCP BENCHMARKS

The evaluation of MCP systems is still in its early stages and continues to evolve rapidly. Among existing efforts, MCPBench (Luo et al., 2025) represents one of the first benchmarks, mainly comparing MCP tools with traditional API-based tools. Building on this, MCP-RADAR (Gao et al., 2025) broadens the scope by introducing a multi-dimensional framework that evaluates efficiency, accuracy, and robustness. More recently, MCPEval (Liu et al., 2025) has advanced the field with a fine-grained framework capable of automatically generating queries to assess MCP server performance.

Despite these advances, current benchmarks share a key limitation: they typically evaluate only small-scale MCP servers (around 10 servers), which fail to capture real-world settings where agents must operate in large, dynamic MCP ecosystems. To address this gap, we introduce a large-scale MCP toolset and systematically examine agent capabilities in accomplishing daily tasks through tool use.

Recent work, such as RAG-MCP (Gan & Sun, 2025), MCPZero (Fei et al., 2025), and ScaleMCP (Lumer et al., 2025) has begun exploring retrieval over large-scale MCP toolsets. However, these approaches are constrained by rigid pipelines that lack adaptability in tool invocation and error recovery. Moreover, ScaleMCP depends on a manually constructed toolset with limited functional diversity, reducing its applicability to broader real-world scenarios.

## 6 CONCLUSION

We presented LiveMCPBench, a benchmark consisting of 95 daily tasks for evaluating tool-use agents in large-scale MCP ecosystems. Alongside, we released LiveMCPTool, a ready-to-use collection of 527 tools, and proposed LiveMCPEval, an automatic evaluation framework that handles task dynamism and solution diversity. We further built the MCP Copilot Agent to study tool retrieval and composition in large-scale MCP settings. Our experiments across ten frontier models highlight persistent limitations: tool retrieval remains the primary bottleneck, while effective tool composition proves equally critical. These findings underscore the need for future systems that jointly advance tool retrieval and foster robust tool composition capabilities.

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

## A    LIMITATIONS

While LiveMCPBench represents a comprehensive benchmarking framework, we acknowledge several limitations in its design and evaluation methodology:

**Dependence on LLM evaluation.**    The LiveMCPEval component relies heavily on LLM-based evaluation. Although we have validated the accuracy through human experiments, potential model biases may still influence the results. To mitigate this concern, we conducted case studies analyzing model judgment failure cases, which helps improve the robustness of our evaluation framework.

**Evaluation Assumptions.**    Our assessment framework operates under the assumption that agent behavior trajectories and tool descriptions sufficiently reflect task performance, without explicitly verifying the final environmental impact. While this assumption holds in most cases, expanding the toolset could introduce inconsistencies between actual tool effects and their descriptions, potentially compromising evaluation reliability. To address this, we rigorously inspect the quality of LiveMCPTool to minimize such discrepancies.

**Server Reliability and Maintenance.**    While our framework highlights the "plug-and-play" nature of LiveMcpTool, we acknowledge that the current work does not explicitly address long-term aspects such as security, versioning, and maintenance of deployed servers. In practice, sustainable applicability requires systematic guidelines for server updates, monitoring, and health curation. To ensure reproducibility of our reported results, we provide a fixed and validated toolset version that has undergone security checks, which guarantees a safe and consistent experimental environment. Beyond this controlled release, we view long-term sustainability as an orthogonal but essential research direction: future iterations of LiveMCPBench could incorporate auditing modules or best-practice recommendations (e.g., periodic health checks, semantic versioning, and community-curated registries) to further ensure secure and reliable deployment.

## B    ETHICAL CONSIDERATIONS

The advent of large-scale multi-tool retrieving and calling agents promises to revolutionize traditional UI-based interaction paradigms by shifting from complex message retrieval or manual UI operations to automated tool invocation. This transition holds significant potential to reduce usability barriers, enhance operational efficiency, and accelerate progress toward Artificial General Intelligence (AGI). Furthermore, such systems can augment the capabilities of smaller models through automated tool construction by larger models. For instance, when faced with tasks beyond their native competence (e.g., complex code generation), smaller models can leverage tools dynamically encapsulated by larger models through MCP interfaces.

However, alongside these benefits, our framework introduces potential risks that warrant careful consideration. Malicious actors could exploit the system by disguising harmful or unsafe tools through misleading descriptions, potentially inducing models to execute dangerous operations. Such misuse may lead to information security breaches or financial losses. Additionally, erroneous tool invocation by the model—such as unintended deletion of local files—could cause significant losses, underscoring the need for robust safeguards in tool validation and execution monitoring.

## C    LLM USAGE

We used LLMs, specifically OpenAI's GPT-based tools, only to check grammar, improve readability, and polish draft wording. All conceptual, technical, and scientific contributions are entirely the authors' own.

## D    REPRODUCIBILITY STATEMENT

We provide in the supplementary material all the code, agent trajectories, and human annotations used in our experiments. In addition, we include the complete set of tasks and the entire LiveMCP-Tool.

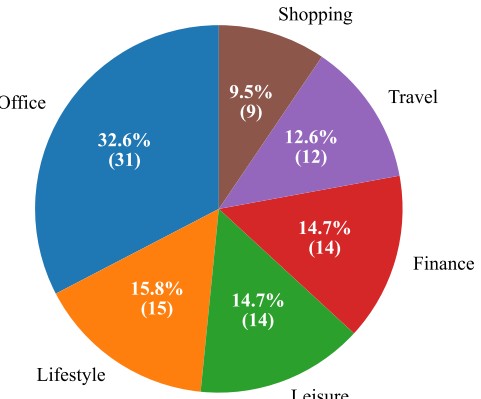

Figure 7: Task distribution in LiveMCPBench: a comprehensive benchmark comprising 95 daily tasks across 6 distinct domains.

Furthermore, Appendix G details the computational resources we utilized, the access methods for private model APIs, and the configuration of the tool retrieval system.

# E DETAILS OF TASK CONSTRUCTION

## E.1 TASK STATISTICS

The task statistics of LiveMCPBench are illustrated in Figure 7. LiveMCPBench comprises six categories of tasks, each designed to reflect common real-life scenarios:

1. **Office.** This category represents typical office-related tasks, primarily involving reading and writing documents in Word, Excel, and PowerPoint.

2. **Lifestyle.** These tasks pertain to daily routines, such as retrieving news updates or querying the latest arXiv papers.

3. **Leisure.** This category encompasses entertainment-oriented tasks, including fetching gaming news, obtaining specific game-related information, or retrieving details about museums.

4. **Finance.** Tasks in this category focus on personal financial management, such as checking stock prices, analyzing market trends, or obtaining cryptocurrency valuations.

5. **Travel.** This category includes tasks related to personal travel, such as route planning, hotel searches, and ticket inquiries.

6. **Shopping.** These tasks revolve around personal shopping activities, including product information retrieval and recommendations.

Importantly, the retained 95 tasks were selected not for sheer quantity, but for domain coverage, generality, and compositional complexity. For instance, beyond single-step retrieval, many tasks involve multi-hop reasoning (e.g., analyzing market data followed by producing a spreadsheet report) or cross-domain integration (e.g., combining travel planning with lifestyle queries). Compared to prior agent benchmarks, LiveMCPBench thus prioritizes both breadth of domains and depth of task composition, providing a balanced yet realistic testbed despite the smaller absolute number of tasks relative to very large-scale static datasets.

**Annotator MCP Usage.** To accomplish the benchmark tasks, annotators utilized 55 out of the 70 available servers (78.57%) and 150 out of the 527 available tools (28.46%). This indicates that while not all resources were required, a substantial and diverse subset of servers and tools was actively engaged, reflecting realistic large-scale MCP tool usage. In practice, many MCP servers contain noisy or redundant tools that are not essential for task completion, and our benchmark naturally captures this characteristic by relying on only the most relevant subset.

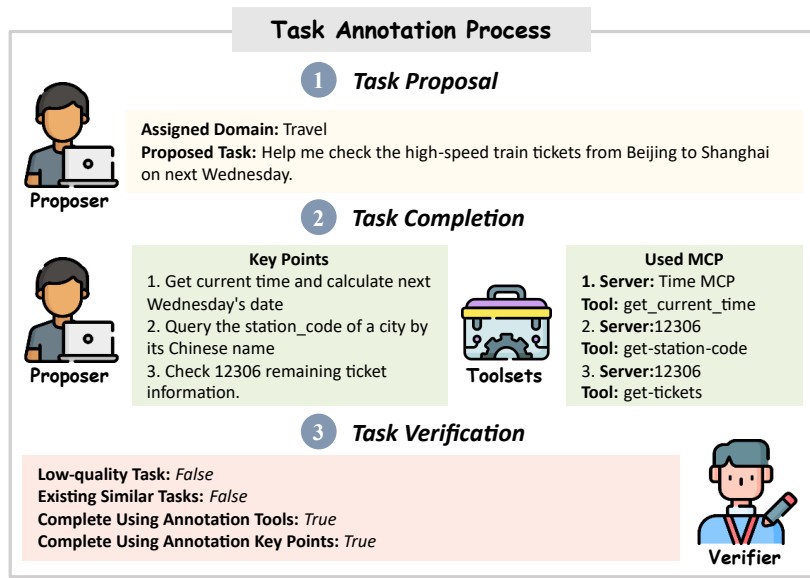

Figure 8: Example of the task annotation process.

### E.2 ANNOTATION PRINCIPLES

LiveMCPBench focuses on leveraging a large-scale MCP toolset to accomplish complex tasks. To ensure high-quality task construction, we employ two groups of annotators involving *proposers* and *verifiers* (see Figure 8). All annotators first freely explore the MCP toolset, including tool descriptions and real-world calls, to gain familiarity with its functionalities.

First, *Proposers* are randomly assigned a scenario and instructed to formulate tasks adhering to the following principles:

1. **Real-World Relevance.** Tasks must reflect realistic needs within the given scenario.
2. **Temporal Dynamics.** Tasks should be time-sensitive, requiring real-time information retrieval from tools rather than relying solely on static internal knowledge.
3. **Tool Diversity.** Tasks should necessitate the integration of multiple tools, avoiding cases where a single tool suffices for completion.

After proposing a task, the *proposers* try to complete it using the MCP toolset, documenting the required tools and key points.

Once all tasks are collected, *verifiers* manually consolidate similar tasks to prevent redundancy. Additionally, they rigorously assess task feasibility and execution quality to maintain high standards in the benchmark.

## F DETAILS OF LIVEMCPTOOL COLLECTION

### F.1 TOOLSET STATISTICS

The statistics of LiveMCPTool's tools and servers are illustrated in Figures 9-10. The collected toolset is categorized into eight distinct classes:

1. **Discovery.** This category encompasses tools for information gathering and retrieval, such as search engines and news aggregators.
2. **Visualization.** Tools in this category facilitate data or concept visualization, including bar chart plotting and mind map creation.

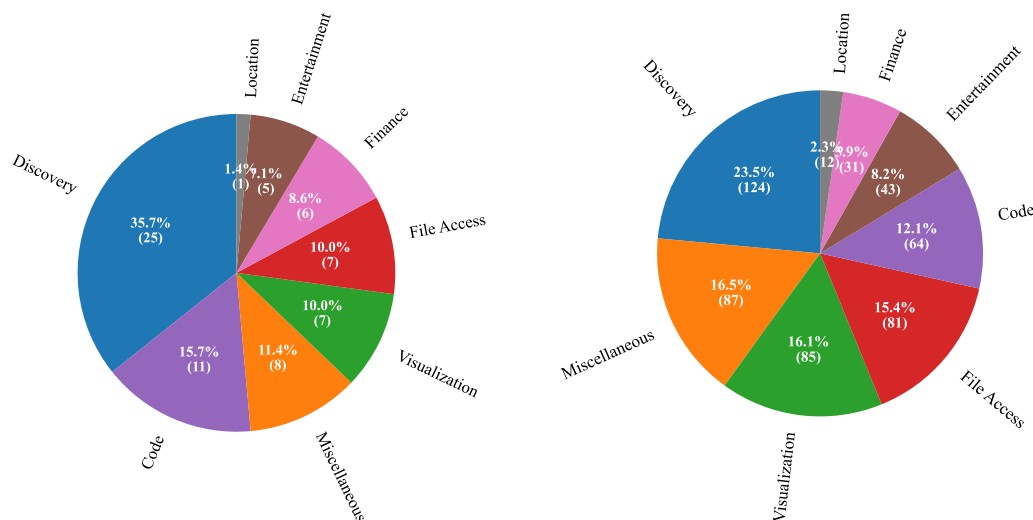

Figure 9: Distribution of servers in LiveM-CPTool, categorized into 8 distinct types (70 servers).

Figure 10: Distribution of tools in LiveMCP-Tool, categorized into 8 distinct types (527 tools).

3. **File Access.** This class comprises tools for local file operations, such as reading Word, Excel, or PowerPoint files, as well as executing command-line instructions.

4. **Code.** These are programming-related tools, such as those providing the latest AntV documentation and sample code.

5. **Entertainment.** This category includes recreational tools, such as those for retrieving Yu-Gi-Oh card information or League of Legends game data.

6. **Finance.** Financial tools fall under this class, including those for fetching real-time stock prices or cryptocurrency market data.

7. **Location.** This category consists of map-based services, such as navigation systems and points-of-interest discovery tools.

8. **Miscellaneous.** This catch-all category accommodates tools not fitting the above classifications, such as calculators and local memory utilities.

**Tool Verification for LiveMCPTool.** To address concerns regarding the dynamism of the LiveM-CPTool, we conducted a systematic verification of all collected tool implementations. For tools that require online information (e.g., weather, news, or stock data), we confirmed that they indeed retrieve real data from online data sources rather than relying on simulated responses or pre-cached static datasets. This validation ensures that the server behaviors used in our benchmark faithfully reflect dynamic, real-world conditions, thereby supporting our claim of genuine runtime dynamism.

## G  IMPLEMENTATION DETAILS

### G.1  COMPUTING RESOURCES AND PRIVATE MODELS

In our experiments, we deployed two models: Qwen2.5-72B-Instruct and Qwen3-Embedding-0.6B. The computational infrastructure consisted of a Linux server (Ubuntu 22.04) with 4 NVIDIA A800-80G GPUs and 1TB of memory.

We accessed the following proprietary models through their respective platforms:

- **OpenRouter**: GPT-4.1, GPT-4.1-Mini, DeepSeek-R1-0528, DeepSeek-V3-0324, Qwen3-235B-A22B and Qwen3-32B.

Table 4: Ablation on tool-return size ($k$) and embedding model, using Claude-Sonnet-4 as the agent model and DeepSeek-V3 for evaluation. * indicates statistical significance at the 0.05 level.

| Settings | Overall (%) | McNemar p-value |
|---|---|---|
| Qwen3-Embedding-0.6B + $k$=5 *(main)* | 78.95 | - |
| *Change $k$* | | |
| Qwen3-Embedding-0.6B + $k$=1 | 64.21 | 0.02* |
| Qwen3-Embedding-0.6B + $k$=10 | 78.95 | 1.00 |
| *Change Embedding Model* | | |
| BGE-M3 + $k$=5 | 76.84 | 0.84 |

- **Anthropic Console**: Claude-Opus-4-20250514, Claude-Sonnet-4-20250514.

- **Google AI Studio**: Gemini-2.5-Pro.

To address suboptimal greedy decoding in certain reasoning models, we implemented a uniform temperature parameter of 0.7 across all experiments. This configuration introduces controlled stochasticity while maintaining result reliability for long-horizon tasks, as we observed that sporadic randomness has a negligible cumulative impact on aggregate performance. We compute the total expenditure based on the model prices provided by OpenRouter as of July 27, 2025, accounting for both input and output tokens.

## G.2 Tool Retrieval Configuration

To ensure consistent retrieval performance across experiments, we adopt a standardized configuration. Specifically, we employ `Qwen2.5-72B-Instruct` to generate server-level tool descriptions and `Qwen3-Embedding-0.6B` to obtain dense embeddings. All retrieval module hyperparameters are kept identical across experimental settings to eliminate variance introduced by this component. Furthermore, in every experiment, the retrieval system deterministically returns the top-5 tools most relevant to the query, ensuring a controlled and comparable evaluation setup.

## G.3 Impact of Tool Retrieval Configuration

To evaluate how the retriever configuration affects the performance of the MCP Copilot Agent, we conduct an ablation study on two factors: the number of returned tools and the choice of vector embedding model. Results are summarized in Table 4. For all paired comparisons against the main setting, we additionally apply an exact McNemar test on per-instance success outcomes, which is appropriate for our matched binary predictions and avoids asymptotic assumptions. The resulting $p$-values are included in the table, providing evidence for the robustness of our primary hyperparameter choice.

Regarding $k$, increasing the tool-return size from 1 to 5 yields a substantial improvement in task success rate (from 64.21% to 78.95%), and this gain is statistically significant under the exact McNemar test ($p = 0.02 < 0.05$). In contrast, further increasing $k$ to 10 produces no measurable benefit (both settings achieve 78.95%, with $p = 1.00$), indicating that the main configuration is already near-optimal with respect to $k$. We attribute this saturation to the retriever's limited ability to surface feasible tools, rather than to an insufficient candidate pool.

For the embedding model, substituting `Qwen3-Embedding-0.6B` with `BGE-M3` leads to only a minor decrease in performance (from 78.95% to 76.84%), and the difference is not significant by the exact McNemar test ($p = 0.84$). Taken together, these statistically grounded comparisons suggest that our main experimental setting is stable to reasonable variations in both $k$ and embedding choice, and further support the conclusion that the dominant bottleneck lies in the current tool-retrieval methodology itself rather than in a specific embedding model or hyperparameter.

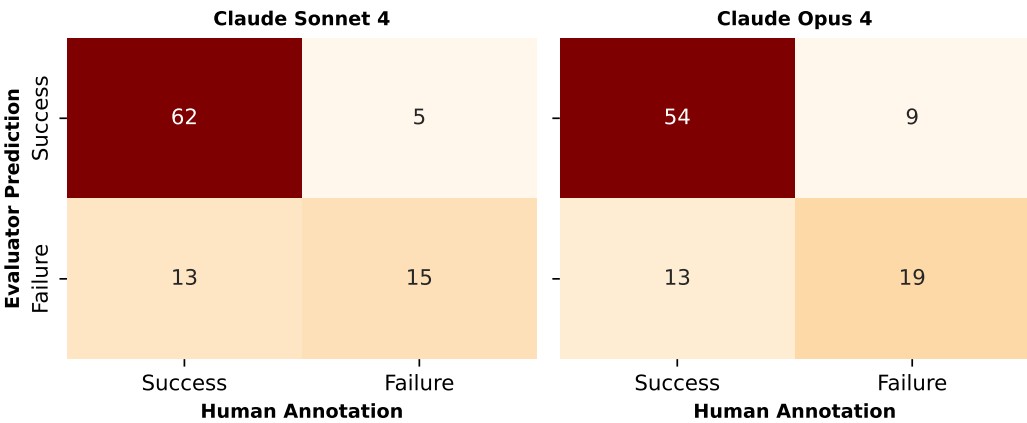

Figure 11: Confusion matrices comparing model-based evaluation (DeepSeek-v3) against human annotators when judging trajectories of the Claude family.

### G.4 ADDITIONAL DISCUSSION ON TOKEN USAGE AND MODEL BEHAVIOR

Table 3 reports the average number of steps and total token consumption for each model. As expected, token usage increases with the number of steps. However, we observe that Claude-Sonnet-4 and Claude-Opus-4 consume substantially more tokens ($\sim$6–7M) compared to the other models ($\sim$2–3M). Beyond their higher average step counts (Claude: $\sim$22 steps; others: $\sim$10 steps), their more aggressive tool-use behavior also contributes significantly to the increased token usage.

For example, Claude-Sonnet-4 requires an average of 20.09 steps and consumes 6.70M tokens, whereas Qwen3-235B requires 16.76 steps and consumes only 3.73M tokens. This large gap is largely explained by their tool-use patterns: Claude-Sonnet-4 invokes an average of 2.71 tools per task, while Qwen3-235B uses only 1.59. These findings suggest that more aggressive tool exploration leads to substantially higher token consumption, due to factors such as more exploratory trials and error-recovery attempts.

## H EVALUATION ANALYSIS

### H.1 EVALUATION METRICS AND JUDGE MODEL SELECTION

In LiveMCPBench, we primarily adopt the single-run success rate as the main comparison metric. However, depending on different evaluation needs, alternative metrics may be more appropriate. For example, the *pass@k* metric (success in at least one out of $k$ independent samples) can be used to assess an agent's ability to discover diverse solution paths. Conversely, the *passˆk* metric (success in all $k$ independent samples) evaluates the agent's consistency and reliability.

For the judge model used in LIVEMCPEVAL, we recommend employing `DeepSeek-V3-0324` to obtain results comparable to those reported in this paper. Nevertheless, other models with higher human agreement or more robust evaluation frameworks may also be used, particularly by leveraging our annotated trajectories based on the Claude family.

### H.2 EVALUATION ERROR ANALYSIS

Figure 11 presents the confusion matrices between the model-based evaluator (DeepSeek-v3) and human annotators for the Claude family of models. The evaluation of claude-sonnet-4 achieved an F1 score of 87.32, while claude-opus-4 reached 83.08, underscoring the relatively high reliability of model-based evaluation. To better understand the observed discrepancies between human judgments and automatic evaluation, we conducted a detailed error analysis.

For clarity and mutual exclusiveness, we categorize evaluation errors into the following four types:

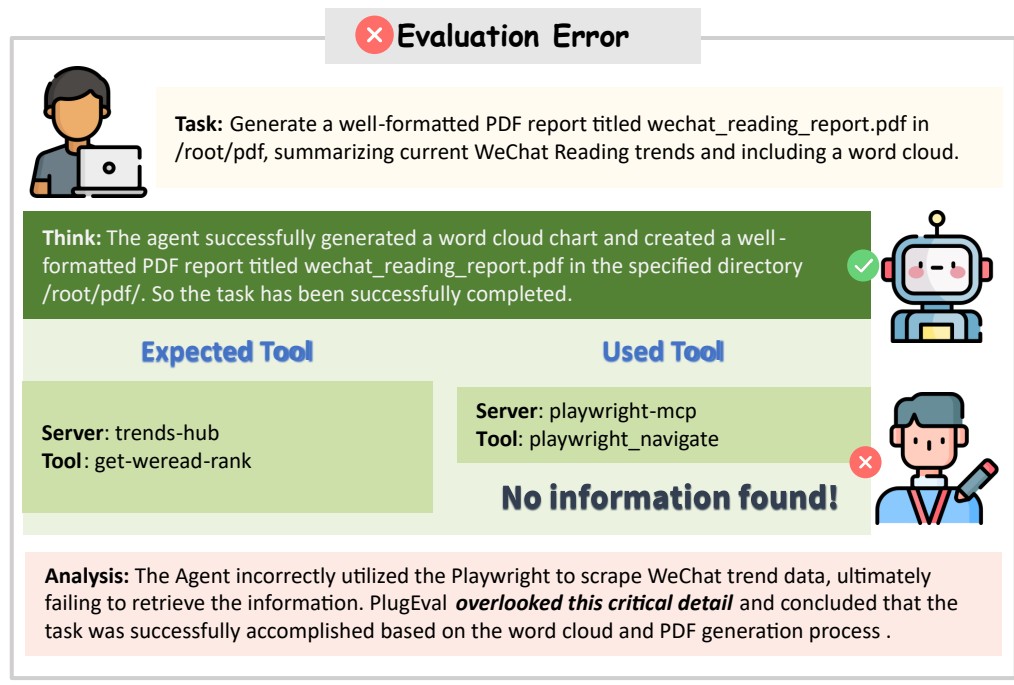

Figure 12: An illustrative case of evaluation failure in LiveMCPEval. The evaluator model erroneously concluded the task was successful based solely on the agent's file creation action, while failing to recognize that the agent did not actually acquire the required information.

- **Output Completeness Error (30%)**: Cases where the evaluator disagreed with humans on whether the model output provided a fully correct and final solution, versus only partial or intermediate steps.

- **Hallucinated Completion (45%)**: Cases where the agent incorrectly assumed completion of a task or access to unavailable information, leading the evaluator to mistakenly align with this hallucination.

- **Granularity Mismatch (17.5%)**: Cases where the evaluator and humans diverged due to differences in tolerance for detail, such as penalizing superficial formatting inconsistencies or overlooking missing justifications.

- **Others (7.5%)**: Residual errors, primarily attributable to human annotation mistakes rather than systematic evaluator weaknesses.

## H.3 EVALUATOR RELIABILITY

Although the reliability of LiveMCPEval has been preliminarily validated through human agreement rates, further in-depth analysis of its applicability and potential improvements is crucial for the long-term sustainability of the evaluation process.

### H.3.1 EVALUATOR BIAS

The potential impact of evaluator bias on the reliability of LLM-as-Judge has raised concerns about the robustness of such evaluation systems.

A potential solution is majority voting. To assess the effectiveness of this approach, we used all 10 models from the main experiment as evaluators to assess the trajectory of Claude-Sonnet-4. The default evaluator, deepseek-v3, has a human agreement rate of 81.05%, while the weakest evaluator, deepseek-r1, has a human agreement rate of 58.95%. We considered results with at least a half vote as predictions, yielding a human agreement rate of **77.89%**. This rate is slightly lower than that of

Table 5: Reevaluation of main experiment results using Qwen 2.5 72B. Relative rankings remain virtually unchanged.

| Model | Overall (%) | Origin Rank |
|---|---|---|
| Claude-Sonnet-4 | 77.89 | 1 |
| Claude-Opus-4 | 64.21 | 2 |
| Qwen3-235B | 40.00 | 3 |
| Qwen2.5-72B | 38.95 | 6 |
| Deepseek-R1 | 37.89 | 4 |
| GPT-4.1-Mini | 31.58 | 5 |
| Deepseek-V3 | 31.58 | 7 |
| Gemini-2.5-Pro | 29.79 | 8 |
| GPT-4.1 | 27.37 | 9 |
| Qwen3-32B | 25.26 | 10 |

the default evaluator but significantly higher than the weakest evaluator. To further explore the upper performance limit of multiple models, we calculated the maximum human agreement rate (where any model output agreeing with human results is considered correct) to be **97.89%**. These findings suggest that when the performance of individual evaluators is uncertain, using a multi-model voting approach is a more robust choice.

To further verify whether the specific model bias of our default deepseek-v3 evaluator affects the relative ranking of performance, we also used **Qwen2.5-72B-Instruct as an evaluator** and reassessed all trajectories. The experimental results, shown in Table 5, indicate that the relative rankings remained almost unchanged, except for Qwen2.5-72B's inherent preference for its own outputs, which could be due to the model's difficulty in identifying errors in its own trajectory.

To quantitatively examine the consistency between the two evaluator, we further computed Kendall's $\tau$-b, which measures the ordinal association between two ranked lists while accounting for ties. The result ($\tau$-**b = 0.8864, p = 0.0004**) indicates a strong and statistically significant agreement between the rankings. In other words, although individual evaluators may exhibit model-specific preferences, their induced orderings over trajectories are largely aligned.

### H.3.2  TOOL-USE HALLUCINATION

Tool-use hallucination is a major source of evaluation failure. Because the evaluator judges task completion solely from the agent's trajectory, hallucinated tool calls or fabricated tool outputs can mislead the evaluator. Therefore, a careful analysis and practical mitigation strategy are necessary.

Conventional rule-based, post-hoc verification is difficult to scale in a large and dynamic MCP environment. Different tool combinations (e.g., multiple search engines in the MCP ecosystem) can achieve equivalent functionality, making it hard to encode capability-equivalence checks with fixed rules. Likewise, answer–based verification is brittle under dynamic tasks such as daily news summarization. As a result, reliable evaluation protocols that are robust to hallucinations are challenging to deploy in real MCP settings.

A promising alternative is to have the evaluator decompose the task into explicit completion criteria and then locate supporting evidence directly from the raw tool outputs. Tool-use hallucinations most often arise when the agent invents intermediate information across multi-step reasoning, and the evaluator fails to notice the fabrication. Evidence-grounded, criterion-wise evaluation can substantially reduce this risk. This can be viewed as an extension of our key-points–based evaluation: each key point is verified independently against the tool sequence.

To quantify the prevalence of tool-use hallucination, we analyzed 814 tool invocations produced by Claude-Sonnet-4. Only **9.00%** involved hallucination, indicating that such errors are relatively infrequent. To assess their impact on evaluation, we measured disagreement between the DeepSeek-V3 evaluator and human judgments due to tool-use hallucination across Claude-family models. The inconsistency rate was only **1.6%**. Among 11 observed hallucination-induced errors, 8 were detected

by the evaluator, yielding a **72.7%** detection rate. Thus, the current protocol already catches most such failures, and the remaining undetected cases place a very small upper bound on evaluation.

Finally, we tested whether tool-use hallucinations are reliably detected across evaluators. We compared the distribution of evaluation errors when using **Qwen2.5-72B** as the evaluator against the default DeepSeek-V3. If detection were unstable, the error-type distribution would shift substantially. Instead, the distribution remained similar: Completeness Error (33.3%), Hallucinated Completion (46.2%), Granularity Mismatch (15.4%), and Others (5.1%). This close match suggests that hallucination-related failures are captured consistently rather than introducing evaluator-dependent bias. Therefore, tool-use hallucinations are unlikely to materially distort our evaluation results.

### H.4 CASE STUDY: ERROR EVALUATION EXAMPLES

To illustrate cases where evaluator judgments diverge from human assessments, we conducted a case study on Claude-Sonnet-4 trajectories evaluated by DeepSeek-V3, presenting a representative example in Figure 12.

In this instance, the evaluator failed to recognize that the agent did not actually acquire the correct information, despite successfully creating the required file. The evaluator erroneously concluded task completion based solely on the file creation process. This case highlights a potential limitation of LiveMCPEval: the system's tendency to overlook critical details when processing excessively lengthy and complex trajectories.

We propose that this long-range evaluation challenge could be addressed by modifying existing evaluation frameworks to incorporate dynamic agent-based assessment of each trajectory step. However, such an approach would significantly compromise evaluation efficiency. While our current evaluation method achieves satisfactory human agreement rates, this particular issue warrants further in-depth investigation.

### H.5 HUMAN KEY POINTS: PRODUCTION AND VALIDATION

**What constitutes a key point.** A *key point* is an actionable, minimal step that contributes to solving the task using the MCP toolset. Each key point must (i) be **atomic** (one tool-mediated action per point), (ii) be **anchored** to a concrete tool or observable signal, (iii) be **auditable** from logs (inputs/outputs). Proposers write key points while completing the task, following the principles in Section E.

**Normalization and consolidation.** After collection, *verifiers* normalize key points into a canonical set by enforcing: (1) imperative voice; (2) one action per point; (3) explicit dependencies (e.g., "after obtaining X from `Tool-A`, call `Tool-B` with Y"); (4) de-duplication of paraphrases and overlapping sub-steps. When multiple proposers describe similar operations with different granularity, verifiers keep the coarsest level that remains auditable and does not obscure required tool interactions.

### H.6 CASE STUDY: HUMAN AND LLM KEY POINTS EXAMPLES

To analyze the differences between LLM-generated key points and human-annotated key points, we conducted a comparative study between key points generated by Deepseek-V3 and those manually annotated by humans. The comparison results are presented in Figure 13.

Our analysis reveals that while the ordering of key points differs between human and LLM-generated outputs, both consistently capture similar critical steps. This observation suggests the practical applicability of LLM-generated key points in evaluation tasks. Importantly, our findings indicate that LLM-generated key points can serve as a reliable alternative for robust evaluation in scenarios where human annotations are unavailable.

## I CASE STUDY: ERROR EXAMPLES

Figures 14-17 present concrete examples of four distinct error types: *Query Error (13.33%)*, *Retrieve Error (50.00%)*, *Tool Error (18.33%)*, and *Other Error (18.33%)*.

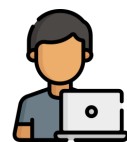
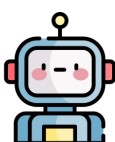
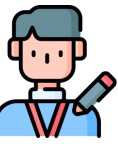

**Human and LLM Key Points**

**Task:** Generate a well-formatted PDF report titled wechat_reading_report.pdf in /root/pdf, summarizing current WeChat Reading trends and including a word cloud.

1. Generate a PDF report.
2. Title the report "wechat_reading_report.pdf".
3. Save the report in "/root/pdf".
4. Summarize current WeChat Reading trends.
5. Include a word cloud.

1. Getting the current trends in WeChat Reading, focusing on popular novels, best-selling books, new releases, and various literary genres.
2. Generate a word cloud chart to visualize the most frequently mentioned words or themes in the top-ranked books.
3. Create a word document.
4. Write the word document.
5. Convert word document to pdf.

Figure 13: Comparison of key points in DeepSeek-V3 and human annotations: Similar content despite different ordering.

Broadly speaking, *Query* and *Other* errors primarily highlight design flaws in the agent's architecture—specifically, whether the agent incorporates sufficient mechanisms to ensure task completion. In contrast, *Tool* errors are more closely tied to the capabilities of the LLM itself, particularly its ability to accurately process tool parameters and descriptions while maintaining nuanced contextual understanding. *Retrieve* errors, on the other hand, largely reflect the limitations of the tool retrieval system, testing its effectiveness in identifying relevant tools based on the server-tool description.

## J    PROMPTS

### J.1    MCP COPILOT AGENT PROMPT

---
**Prompt for MCP Copilot Agent**

You are an agent designed to assist users with daily tasks by using external tools. You have access to two tools: a retrieval tool and an execution tool. The retrieval tool allows you to search a large toolset for relevant tools, and the execution tool lets you invoke the tools you retrieved. Whenever possible, you should use these tools to get accurate, up-to-date information and to perform file operations.
Note that you can only response to user once, so you should try to provide a complete answer in your response.

- - - - - - - - - - - - - - - - - - - - - - - - - - - - - - - - - - - - - - - - -

```
Task
Tool
```
: mcp-copilot (with *route* and *excute* tool)

---

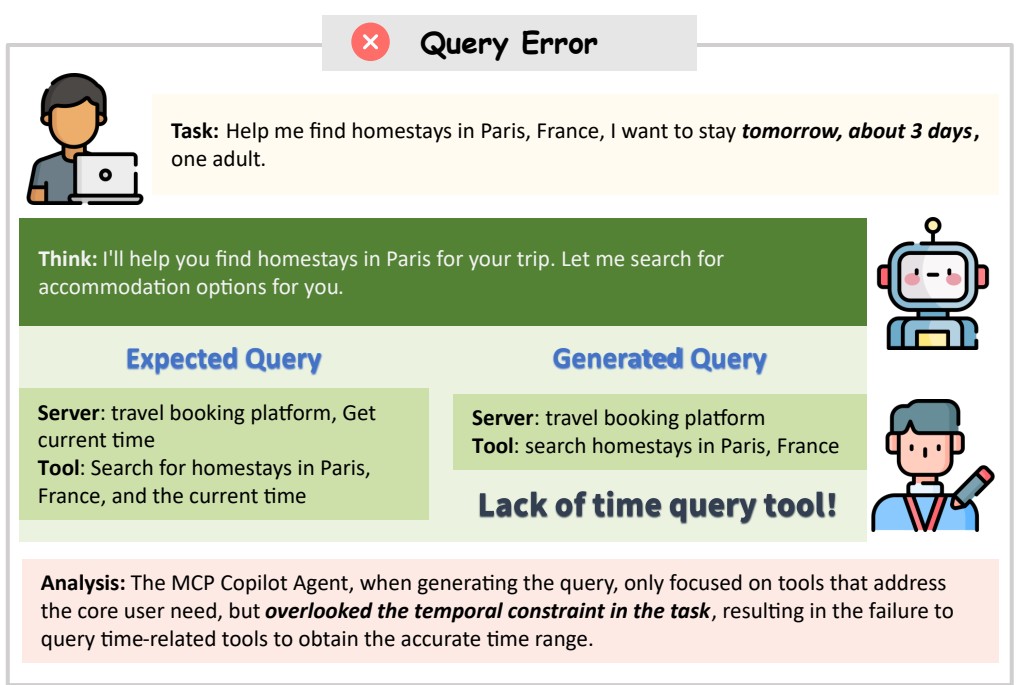

Figure 14: An illustration of **Query Error**: Discrepancy between the agent-generated query and the task's required competencies.

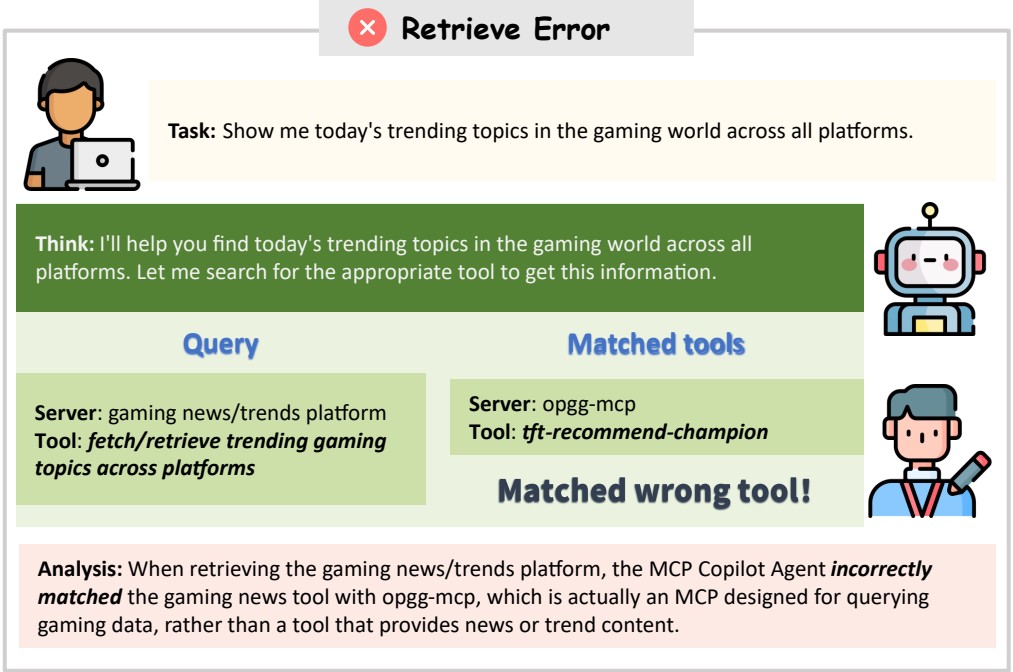

Figure 15: An illustration of **Retrieve Error**: The retrieve system incorrectly identifies and returns an inappropriate tool.

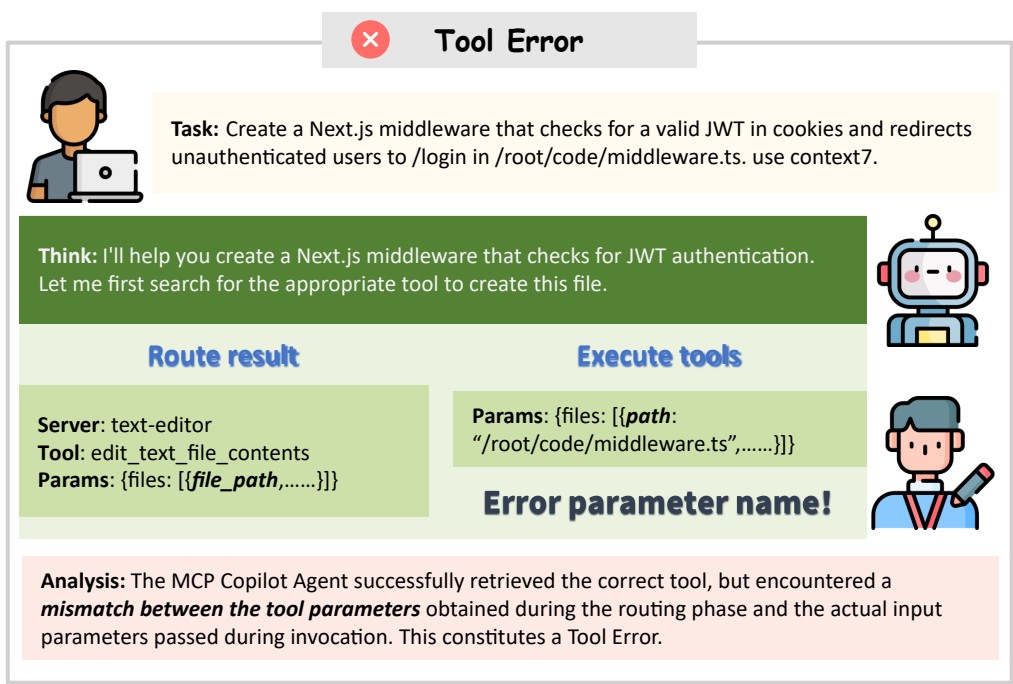

Figure 16: An illustration of **Tool Error**: The LLM misapplication with improper tool parameters.

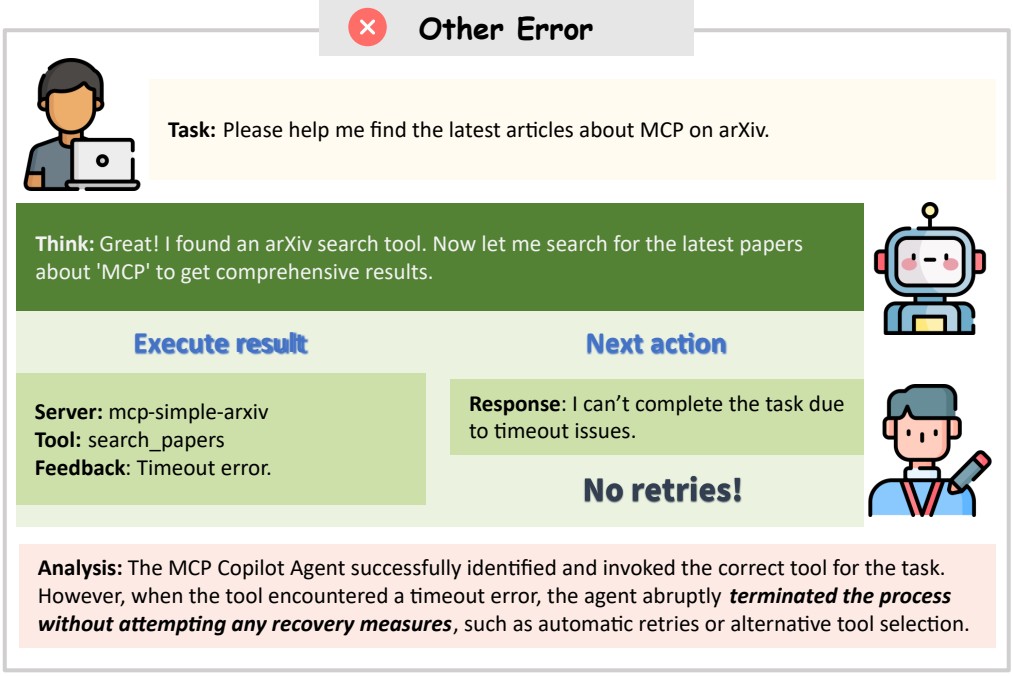

Figure 17: An illustration of **Other Error**: The agent's inadequate response to tool timeout.

**Prompt for *route* tool**

This is a tool used to find MCP servers and tools that can solve user needs When to use this tool:
-When faced with user needs, you (LLM) are unable to solve them on your own and do not have the tools to solve the problem.
-When a user proposes a new task and you (LLM) are unsure which specific tool to use to complete it.
-When the user's request is vague or complex, and feasible tool options need to be explored first.
-This is the first step in executing unknown tasks, known as the "discovery" phase, aimed at finding the correct tool.
**Parameter Description**
Query (string, required): The input query must contain a `<tool_assistant>` tag with server and tool descriptions, for example:
`<tool_assistant>`
server: ... # Platform/permission domain
tool: ... # Operation type + target
`</tool_assistant>`

**Prompt for *excute* tool**

A tool for executing a specific tool on a specific server.Select tools only from the results obtained from the previous route each time.
When to use this tool:
- When using the route tool to route to a specific MCP server and tool
- When the 'execute-tool' fails to execute (up to 3 repetitions).
- When the user's needs and previous needs require the same tool.
Parameters explained:
-server_name: string, required. The name of the server where the target tool is located.
-tool_name: string, required. The name of the target tool to be executed.
-params: dictionary or None, optional. A dictionary containing all parameters that need to be passed to the target tool. This can be omitted if the target tool does not require parameters.

**Prompt for server summary**

You are an expert AI technical writer. Based on the following information about an MCP server, please generate a concise and accurate summary of its core purpose and capabilities.
**Server Name:** `server_name`
**Server Description:** `server_desc`
**Available Tools:** `tool_descriptions`
Please return only the generated summary text, without any additional titles or preambles.

## J.2 LIVEMCPEVAL PROMPT

---

**Prompt for evaluation**

You are an expert in evaluating the performance of a tool-use agent. The agent is designed to help a human user use multi-tools to complete a task. Given the user's task, the agent's final response, key points for task completion, and tool call history, your goal is to determine whether the agent has completed the task and achieved all requirements.
Your response must strictly follow the following evaluation criteria!
*Important Evaluation Criteria*:
1. You must carefully check whether the information (e.g. the coordinates of the addresses) comes from the tool call, if the agent get it from the internal knowledge, it should be considered failed.
2: Some tasks require to create files to be considered successful.
*IMPORTANT*
Format your response into two lines as shown below:
Thoughts: <your thoughts and reasoning process based on double-checking each key points and the evaluation criteria>
Status: "success" or "failure"

- - - - - - - - - - - - - - - - - - - - - - - - - - - - - - - - - - - - - - - - - - - - - - - - - -

User Task: `task`
Key Points: `key_points`
Final Response: `response`
Tool Call History: `tool_calls`
Tool Descriptions: `tool_descriptions`

---

**Prompt for identify key points**

You are an expert tasked with analyzing a given task to identify the key points explicitly stated in the task description.
**Objective**: Carefully analyze the task description and extract the critical elements explicitly mentioned in the task for achieving its goal.
**Instructions**:
1. Read the task description carefully.
2. Identify and extract **key points** directly stated in the task description.
- A **key point** is a critical element, condition, or step explicitly mentioned in the task description.
- Do not infer or add any unstated elements.
**Respond with**:
- **Key Points**: A numbered list of the explicit key points for completing this task, one per line, without explanations or additional details."""

- - - - - - - - - - - - - - - - - - - - - - - - - - - - - - - - - - - - - - - - - - - - - - - - - -

`task`

