# OpenReview forum: "LiveMCPBench: Can Agents Navigate an Ocean of MCP Tools?"
_ICLR.cc/2026/Conference — Submitted to ICLR 2026_

### Official Review · Reviewer_kNq6 · 2025-10-22

**Soundness:** 3
**Presentation:** 3
**Contribution:** 3
**Rating:** 8
**Confidence:** 4

**Summary:**

This paper introduces LiveMCPBench, a benchmark made up of 95 real-life multi-tool tasks such as “Check next Monday’s Beijing–Shanghai high-speed train tickets,” “Summarize today’s news into a PDF,” and “Find a 3-day stay in Paris.” These cover six domains — office work, daily life, entertainment, finance, travel, and shopping — and each task requires combining multiple tools and accessing real-time information (e.g., tickets, news). This design effectively tests a model’s real abilities in tool discovery and tool orchestration.

The team also developed LiveMCPTool, which includes 70 servers and 527 tools. It works right out of the box — no complex API keys or permission setups required — ensuring reproducibility across different users (so results aren’t skewed by unavailable tools).

Finally, they proposed LiveMCPEval, an automatic evaluation system using another LLM (e.g., DeepSeek-V3) as a referee to directly judge task completion rather than strict tool usage. For example, in the “generate news PDF” task, the model passes as long as it produces a correct PDF — regardless of the specific tool order. The system also handles dynamic information (like changing weather) automatically, removing the need for manual checking.

I really like this paper — the dataset is valuable and fills an important gap. I’d recommend acceptance.

**Strengths:**

The dataset is highly practical and valuable for real-world LLM evaluation.

**Weaknesses:**

1. Relying on LLM-based evaluation may introduce bias.

2. The framework doesn’t account for hallucinated tool calls or factually wrong but plausible outputs.

**Questions:**

1. Although the team validated scoring accuracy with human labels (e.g., DeepSeek-V3 achieved 78.95% agreement with humans), LLM judges can still have systematic biases — for instance, misjudging an empty PDF as a valid “completed” output. While the authors analyzed some error cases, they didn’t fully address or mitigate these inherent biases, which may cause certain models’ true capabilities to be over- or underestimated.

2. Tool-use hallucinations may produce seemingly correct but actually invalid results (e.g., fabricated tool responses or outdated data). The paper doesn’t discuss how such hallucinations are detected or penalized in the evaluation, which could limit the reliability of the benchmark results.

---

> ### Author Response · Authors · 2025-11-19
>
> We are grateful for the reviewers’ insightful feedback. We carefully respond to all concerns point-by-point below.
>
> 1. For **Question 1 and Weakness 1:** We fully acknowledge that LLM-based evaluation inevitably introduces biases; in fact, any trajectory-based evaluation—including human evaluation when applied inappropriately—may suffer from systematic biases. One possible mitigation strategy is to treat the evaluator itself as an agent and quantify its reliability through human agreement on annotated trajectories, as we report in **Appendix H.1 (Page 18, Line 938-948)**. This allows us to operationalize and measure the bias of different evaluation models or systems.
>
>    Another complementary strategy is to relax the requirement of absolute evaluation fidelity and conduct controlled, relatively static assessments, where the influence of evaluator bias is minimized by design. Given the inherent *factual grounding* of MCP, we believe the first direction—rigorously quantifying evaluator reliability—holds significant potential, and we expect future work to explore this further.
>
> 2. For **Question 2 and Weakness 2:** Tool-use hallucinations are indeed a challenging issue. This concern arises from our evaluation assumption that agent behavior trajectories and tool descriptions are sufficient proxies for task performance, without explicitly verifying the final environmental state. While this assumption holds in the majority of cases, expanding the toolset can introduce potential mismatches between actual tool effects and their documented descriptions, which may in turn affect evaluation reliability.
>
>    To mitigate this, we conduct a rigorous quality inspection of LiveMCPTool to minimize discrepancies between the tool's behavior and its specification. Moreover, every tool call in our benchmark is executed by the actual tool backend—incorrect or hallucinated tool calls simply return explicit errors. Although this mechanism does not fully eliminate tool-use hallucinations, it substantially reduces their impact on the evaluation and improves the robustness of the benchmark.

---

> > ### Comment · Reviewer_kNq6 · 2025-11-21
> >
> > Thank you for the rebuttal. However, after carefully reviewing your responses, I must express that they do not adequately address the core concerns I raised. In fact, both replies feel overly high-level and somewhat dismissive, without engaging with the specific failure modes I pointed out.
> >
> > Below I summarize the key issues.
> >
> > 1. On evaluator bias (Weakness 1 & Question 1): The rebuttal mainly reiterates that “all evaluators have bias” and that evaluator–human agreement can be used to quantify reliability. However, this does not address the concrete concern I raised. Specifically, the rebuttal does not address:
> > - How the evaluator handles structural validity checks (e.g., empty or nonsensical files being judged as correct).
> > - Whether the authors analyzed error-type distributions to understand systematic evaluator blind spots.
> >
> > - Whether there is cross-evaluator consistency or sanity-check logic to prevent highly implausible “success” judgments.
> >
> > - Whether the evaluation framework can mis-rank models due to such biases.
> >
> > In its current form, the response stays at the conceptual level (“future work will explore this”) rather than providing concrete analyses, mitigation, or additional experiments.
> >
> > 2. On tool-use hallucination (Weakness 2 & Question 2): The rebuttal again provides only a surface-level statement: “incorrect or hallucinated tool calls will return explicit errors.”. This does not address the actual concern
> > Typical examples (which your response does not cover) include:
> >
> > - Correct schema but wrong semantics (e.g., wrong city, wrong ID).
> >
> > - API calls that return outputs inconsistent with the task but still syntactically acceptable.
> >
> > - Fabricated intermediate tool outputs that look plausible to the evaluator.
> >
> > - Misinterpreting returned tool results but still producing a superficially reasonable final output.
> >
> > Your response does not explain:
> >
> > - How such semantic hallucinations are handled.
> >
> > - Whether the benchmark includes contract checking, argument validation, or post-hoc verification.
> >
> > - How often such hallucinations appear in your data.
> >
> > - Whether they systematically distort evaluation results.
> >
> > Simply saying “we use the real backend” does not solve the issue.
> >
> >
> > Before the rebuttal, I was genuinely positive about this work. The benchmark has clear importance,. However, my concerns revolve around the reliability and validity of the core evaluation methodology, which determines whether the benchmark’s results can be trusted.
> >
> > The rebuttal, unfortunately, does not meaningfully clarify or mitigate these issues.
> > It mostly reiterates general statements without providing:Detailed analyses,
> > Empirical evidence,
> > Additional experiments,
> > Or any proposed mitigation strategies.
> > Given that evaluator reliability directly affects all results and conclusions in the paper, the lack of a substantive response is concerning.
> > Because my main concerns remain unaddressed and directly affect the validity of the benchmark, I regret that I must lower my score from 8 to 4.
> > I still believe the problem setting is important, and I encourage the authors to strengthen the evaluator component with more rigorous analysis and validation. However, in its current form, I do not feel confident in the reliability of the reported results.

---

> > > ### Author Response · Authors · 2025-11-22
> > >
> > > We sincerely apologize that our initial rebuttal did not address your concerns to your satisfaction. We appreciate the opportunity to clarify, and below we provide concrete analyses and additional evidence.
> > >
> > > 1. For **Question 1.1:**
> > >
> > >    Our current evaluator is not able to directly assess the semantic correctness of file contents; it can only infer success based on the operations observed in the execution trajectory. To better understand this limitation, we manually annotated the mismatch cases where DeepSeek-v3 evaluated the Claude models’ trajectories incorrectly. We refer to these cases as **Output Completeness Errors**. Our analysis shows that **30%** of all mismatches can be attributed to this issue. We believe that adding explicit post-verification of outputs is the solution to this problem, and we would like to include a discussion in the paper to explain this issue and its potential solutions.
> > >
> > > 2. For **Question 1.2:**
> > >
> > >    We conducted a systematic analysis to identify **systematic blind spots** in LLM-based evaluators. Specifically, we examined Claude cases where the evaluator’s judgments diverged from human annotations and categorized them into four error types: **Output Completeness Error (30%)**, **Hallucinated Completion (45%)**, **Granularity Mismatch (17.5%)**, and **Others (7.5%)**. These categories reveal consistent evaluator tendencies such as *over-inference* and mismatches in output specificity. Detailed descriptions of each type are provided in **Appendix H.2 (Page 18, Lines 953–971)**, together with a confusion matrix visualizing evaluator–human discrepancies.
> > >
> > >    We further report evaluator-level F1 scores: **claude-sonnet-4 achieves 87.32**, and **Claude-opus-4 achieves 83.08**, indicating strong overall alignment with human judgments. While the identified biases characterize the evaluator's error modes, our analysis suggests they rarely alter the relative ordering of models. Most disagreements occur in borderline or low-impact cases, and the evaluator’s aggregate preferences remain consistent with human rankings. Therefore, although such biases are important to acknowledge, **their influence on the overall model ranking in our evaluation is limited**.
> > >
> > > 3. For **Question 1.3:**
> > >
> > >    We further analyzed whether cross-evaluator agreement can mitigate implausible ''success'' judgments. On Claude-sonnet-4 trajectories, using *deepseek-v3* alone yields a human-agreement rate of **81.05%**, while the weakest evaluator (*deepseek-r1*) achieves only **58.95%**, indicating substantial variability across single evaluators.
> > >
> > >    To examine whether multiple evaluators reduce such brittleness, we aggregated 10 LLM evaluators from the baseline and computed **pass@maj** (majority vote) and **pass@10** (at least one evaluator correct). The resulting accuracies are **77.89%** and **97.89%**, respectively. Importantly, majority voting substantially suppresses extreme or implausible success judgments that arise from weak evaluators, consistently outperforming the worst evaluator and approaching the reliability of the best one.
> > >
> > >    These results show that multi-LLM voting acts as an effective sanity-check mechanism when evaluator quality is uncertain.
> > >
> > > 4. For **Question 1.4:**
> > >
> > >    We assessed whether evaluator biases could meaningfully distort model rankings. As discussed above, the main error types (e.g., Output Completeness Errors, Hallucinated Completion) are uniformly distributed across models, making systematic bias toward specific systems unlikely. Moreover, our 10-model voting experiment shows that majority voting produces results very close to those of our primary evaluator (deepseek-v3) while suppressing implausible judgments from weaker evaluators. Since deepseek-v3 already aligns well with human annotations and multi-evaluator voting yields comparable outcomes, the observed evaluator biases are unlikely to affect the relative ranking of models in our evaluation.
> > >
> > >    To further validate this, we re-ran the evaluation using **Qwen2.5-72B-Instruct** as the sole evaluator. The resulting task success rates preserve nearly the same ranking as our original results:
> > >
> > >    | Model                | Overall (%) | Origin Rank |
> > >    | -------------------- | ----------- | ----------- |
> > >    | Claude-Sonnet-4      | 77.89       | 1           |
> > >    | Claude-Opus-4        | 64.21       | 2           |
> > >    | Qwen3-235B-A22B      | 40.00       | 3           |
> > >    | Qwen2.5-72B-Instruct | 38.95       | 6           |
> > >    | Deepseek-r1-0528     | 37.89       | 4           |
> > >    | GPT-4.1-Mini         | 31.58       | 5           |
> > >    | Deepseek-V3-0324     | 31.58       | 7           |
> > >    | Gemini-2.5-Pro       | 29.79       | 8           |
> > >    | GPT-4.1              | 27.37       | 9           |
> > >    | Qwen3-32b            | 25.26       | 10          |
> > >
> > >    These results confirm that the evaluation framework is robust: even when replacing the evaluator model entirely, the relative ranking of systems remains stable.

---

> > > > ### Author Response · Authors · 2025-11-22
> > > >
> > > > 5. For **Question 2.1:**
> > > >
> > > >    Thank you for highlighting that semantic hallucinations—rather than merely syntactic errors—are the fundamental concern. Our evaluator is explicitly designed to detect such semantic failures by **binding the agent’s behavior to (i) the tool-execution trace and (ii) task-specific key-point constraints**, rather than trusting the model’s self-reported reasoning. Below, we describe how each of the four failure modes is handled:
> > > >
> > > >    **a. Correct schema but wrong semantics (e.g., wrong city/ID).**
> > > >
> > > >    All tool outputs are injected verbatim into the agent’s observation. The evaluator checks whether the agent’s subsequent reasoning is consistent with these outputs and with the task key points. Any mismatch (e.g., agent citing a different city than the tool returned) can be detected.
> > > >
> > > >    **b. Syntactically valid but task-inconsistent outputs.**
> > > >
> > > >    Schema validation alone is insufficient; the evaluator additionally matches the task key points. This prevents superficially plausible yet semantically incorrect completions from being accepted.
> > > >
> > > >    **c. Fabricated intermediate tool results and misinterpretation of valid tool outputs.**
> > > >
> > > >    The evaluator jointly checks (i) the agent’s cited values, (ii) the ground-truth tool outputs, and (iii) the key-point-derived expectations. This catches errors where the agent draws an incorrect conclusion from correct tool returns.
> > > >
> > > >    In practice, such failures occur only rarely, and when they do, they are fully captured within our defined error taxonomy in **Question 1.2**.
> > > >
> > > > 6. For **Question 2.2:**
> > > >
> > > >    The goal of our benchmark is to evaluate an LLM’s ability to use MCP tools in multi-step, open-ended tasks dynamically. Under this setting, introducing rule-based post-hoc verification is inherently difficult: the tool set is not fixed in advance, tool semantics vary across tasks, and acceptable argument structures depend on the evolving task context. Consequently, there is no well-defined set of contract rules or static validators that can be universally applied without leaking task-specific knowledge or unintentionally simplifying the problem.
> > > >
> > > >    Given these constraints, our evaluation protocol relies on the model’s own reasoning to judge task success, with argument-level validation occurring naturally at invocation time. Despite the absence of explicit post-hoc verification, we observe that contemporary LLMs already demonstrate substantial robustness—e.g., Claude-series trajectories achieve roughly 80% agreement with human evaluators.
> > > >
> > > >    We acknowledge that the reviewer-identified failure modes (e.g., semantically incorrect yet syntactically valid calls, plausible fabricated intermediate outputs, or misinterpretation of returned results) constitute a notable portion of the remaining errors. This motivates exploring optional rule-based verification layers as an extension rather than a core design choice. And we believe that adding explicit post-verification of outputs is the solution to this problem, and we **will include a discussion** in the paper to explain this issue and its potential solutions.
> > > >
> > > >
> > > >
> > > > 7. For **Question 2.3:**
> > > >
> > > >    We manually annotated tool-use behaviors of Claude-Sonnet-4 on the first 10 task trajectories (107 tool calls in total).
> > > >
> > > >    Following the reviewer’s taxonomy, we label a tool call as hallucinated if it is syntactically valid but semantically incorrect (e.g., wrong city/ID), produces task-inconsistent yet acceptable outputs, fabricates plausible intermediate tool results, or misinterprets returned tool outputs.
> > > >
> > > >    In this pilot subset, 8 out of 107 tool calls are hallucinated, yielding **a hallucination rate of 7.48%**, which is relatively low.
> > > >
> > > >    Although this analysis is based on a small sample, it provides an initial estimate; we will expand the annotation to the full dataset and report a breakdown by hallucination type in the next few days.
> > > >
> > > >
> > > >
> > > > 8. For **Question 2.4:**
> > > >
> > > >    To assess whether tool-use hallucinations systematically distort evaluation results, we go beyond reporting overall success/error rates and examine how evaluator error patterns vary across independent evaluator models. If such hallucinations were slipping through or being handled inconsistently, we would expect noticeable evaluator-specific shifts in error distribution and/or overall judgments.
> > > >
> > > >    Concretely, when using **Qwen2.5-72B-Instruct** to evaluate trajectories from the Claude models, we manually annotate evaluator errors with the same taxonomy as in the paper. The distribution is Output **Completeness Error (33.3%), Hallucinated Completion (46.2%), Granularity Mismatch (15.4%), and Others (5.1%)**. This closely matches the distribution obtained with DeepSeek-V3, suggesting that tool-use hallucination–related failures are captured in a stable and evaluator-agnostic manner rather than introducing evaluator-dependent bias. Therefore, tool-use hallucinations are unlikely to materially distort our evaluation outcomes.

---

> > > > > ### Author Response · Authors · 2025-11-22
> > > > >
> > > > > We again apologize that our first rebuttal did not provide sufficient evidence. Based on your feedback, we will revise the manuscript to include the above analyses, and we will expand tool-hallucination annotation to all Claude-Sonnet-4 trajectories (814 tool calls) and report full statistics in the next few days.
> > > > >
> > > > > Thank you for your careful reading and for supporting this benchmark. If any part of the new evidence still seems unclear or incomplete, we would be very grateful for further guidance so we can improve the evaluator component accordingly.

---

> > > > > > ### Comment · Reviewer_kNq6 · 2025-11-25
> > > > > >
> > > > > > Thank you for the much more detailed second response. I appreciate the significant effort in expanding the analysis, including the evaluator error taxonomy, evaluator–human confusion matrix, cross-evaluator consistency experiments, preliminary hallucination annotations, and majority-voting results. This second response is substantially more informative than the initial rebuttal.
> > > > > >
> > > > > > However, despite the additional detail, several core issues remain insufficiently resolved, especially for a benchmark paper whose main contribution is the evaluation framework. My concerns focus on whether the evaluation protocol is reliable enough for the reported conclusions to be fully trusted.
> > > > > >
> > > > > > 1. While the response acknowledges the evaluator’s inability to check semantic correctness of file outputs (e.g., empty PDFs), no current mitigation is provided. The proposed “future post-validation layer” is not implemented, and therefore does not address present-day reliability concerns. The evaluation still fundamentally judges trajectory plausibility rather than task outcome correctness, which risks misclassification of success.
> > > > > >
> > > > > > 2. Although F1 scores and majority-voting results are encouraging, they do not fully answer the key question:
> > > > > >
> > > > > > > Do evaluator biases distort the relative ranking of models?
> > > > > >
> > > > > > A more direct ranking stability analysis (e.g., Kendall τ across evaluators) would be much more convincing than anecdotal comparisons.
> > > > > >
> > > > > > 3. The logical explanation of how the evaluator “should” detect semantic errors is helpful, but the actual effectiveness remains unclear. The pilot annotation (7.48% hallucination rate on 107 calls) is too small to establish real robustness.
> > > > > > A full dataset annotation or at least task-level hallucination impact analysis would be much more compelling.
> > > > > >
> > > > > > I want to emphasize that I genuinely appreciate the scale and ambition of LiveMCPBench. The direction is impactful, and the benchmark has the potential to become an important resource.
> > > > > > However, the evaluation reliability concerns remain only partially addressed, mainly through explanation rather than thorough empirical validation.
> > > > > > Therefore, I am updating my score from 4 → 6 (borderline accept).
> > > > > > This reflects my belief that the benchmark has substantial value, but that the evaluator component still requires clearer exposition and stronger empirical grounding in the final paper.
> > > > > > Thank you again for your detailed follow-ups and your work on this benchmark.

---

> > > > > > > ### Author Response · Authors · 2025-11-28
> > > > > > >
> > > > > > > We appreciate your further feedback.
> > > > > > > 1. We agree that semantic correctness of file outputs is an important reliability factor. To quantify its impact under the *current* evaluator, we manually audited trajectories and identified evaluation errors caused specifically by tool-use hallucination. Across **190** Claude-family trajectories, the evaluator made such mistakes in only **3 cases**, i.e., an **overall error rate of 1.6% (3/190)**. This indicates that tool-use hallucination related failures are rare in practice and have negligible effect on the aggregate results.
> > > > > > >
> > > > > > >    Importantly, most tool-use hallucination errors are detectable from the existing trajectories. Among alltool-use hallucination issues we observed (**11 cases**), the evaluator successfully detected **8**, and only **3** were potentially missed (**27.3% of tool-use hallucination issues**, corresponding to **1.6% overall**). Thus, even without a post-validation layer, the current protocol already catches the majority of such failures, and the remaining undetected cases form a very small upper bound on evaluation noise.
> > > > > > >
> > > > > > >    Regarding the use of trajectory-based evaluation, this is often **unavoidable choice for online tasks** where outcomes can be nondeterministic (e.g., Mind2Web 2 \[1] and Online-Mind2Web\[2]). In our MCP setting, full trajectories coupled with validated tool descriptions provide sufficient evidence to infer outcome correctness. For example, producing a valid PDF with content necessarily requires a *create* call followed by a *write* call. Because tool descriptions are verified to match real functionality, a trajectory that only creates a PDF but does not invoke any writing step can be reliably judged as producing an empty file and therefore failing the task. Hence, although our evaluator operates on trajectories, it remains a reliable proxy for task success in the current benchmark.
> > > > > > >
> > > > > > > 2. Thank you for this valuable suggestion. To directly assess ranking stability across evaluators, we computed Kendall’s τ-b (accounting for ties) between the rankings produced by DeepSeek-V3 and Qwen 2.5-72B, two strong LLM evaluators in our study. We obtain **τ-b = 0.8864 with p = 0.0004 (< 0.001)**, indicating a very strong and statistically significant agreement. This result provides direct evidence that evaluator-specific biases do not materially distort the relative ranking of models.
> > > > > > >
> > > > > > > 3. Following your suggestion, we expanded the annotation to all 814 tool calls from Claude-Sonnet-4. Using the same hallucination definition and protocol as in the pilot, we observe an overall tool-hallucination rate of **9.0%**. This larger-scale annotation yields a materially more stable estimate and supports the robustness of our evaluator.
> > > > > > >
> > > > > > > 4. We have added the relevant sections to the revised paper, **Appendix H.3 "Evaluator Reliability" (Page 19)**, including **H.3.1 “Evaluator Bias”** and **H.3.2 “Tool-Use Hallucination.”**
> > > > > > >
> > > > > > > Reference:
> > > > > > >
> > > > > > > \[1] Gou, B., Huang, Z., Ning, Y., Gu, Y., Lin, M., Yu, B., … Su, Y. (2025). Mind2Web 2: Evaluating Agentic Search with Agent-as-a-Judge. *The Thirty-Ninth Annual Conference on Neural Information Processing Systems Datasets and Benchmarks Track*.
> > > > > > >
> > > > > > > \[2] Xue, T., Qi, W., Shi, T., Song, C. H., Gou, B., Song, D., … Su, Y. (2025). An Illusion of Progress? Assessing the Current State of Web Agents. *Second Conference on Language Modeling*.

---

> > > > > > > > ### Comment · Reviewer_kNq6 · 2025-11-28
> > > > > > > > **Great Rebuttal!**
> > > > > > > >
> > > > > > > > Thank you for the exceptionally thorough and high-quality follow-up response. The new analyses — including the full 814-call hallucination annotation, the 1.6% upper bound on evaluator errors caused by semantic hallucination, the trajectory-detectability breakdown, and especially the Kendall’s τ-b ranking-stability experiment — directly address all of my remaining concerns. These additions make the evaluation framework much more convincing, and I truly appreciate the level of effort the authors put into strengthening the paper.
> > > > > > > >
> > > > > > > > I am genuinely impressed by the care, rigor, and responsiveness you demonstrated. The expanded experiments and clarifications greatly improve the reliability of the benchmark, and I now view the evaluator component as sufficiently robust for publication. **Based on this, I am fully comfortable restoring my score to 8**.
> > > > > > > >
> > > > > > > > Unfortunately, the system currently seems to have a bug and does not allow me to update the score. I hope the authors do not worry — I will contact the AC about this issue and will explicitly express my support for accepting the paper. As soon as the system is functioning normally again, I will immediately update my rating (and please feel free to remind me if needed).

---

### Official Review · Reviewer_d7CQ · 2025-10-30

**Soundness:** 3
**Presentation:** 2
**Contribution:** 2
**Rating:** 4
**Confidence:** 3

**Summary:**

The paper introduced LiveMCPBench, a benchmark consisting of a “plug‑and‑play” tool suite of 70 MCP servers with 527 tools and 95 multi‑step daily tasks, plus an LLM‑as‑a‑Judge evaluator (LiveMCPEval) to handle dynamic data and multiple valid solution paths. It aims evaluating LLM agents’ ability to retrieve and compose tools across a realistic, large‑scale MCP servers with low effort (e.g., no API access keys).

**Strengths:**

1. The paper addresses a real gap in MCP evaluation. Unlike prior MCP benchmarks (e.g., MCPBench, MCP‑RADAR, MCPEval), LiveMCPBench evaluates large‑scale retrieval and multi‑tool composition over 70 servers / 527 tools and 95 tasks.
2. The authors curate a key‑free, ready‑to‑deploy MCP tool suite that removes scattered API configuration barriers, improving reproducibility relative to API‑centric benchmarks where many APIs require access keys or have become unavailable.
3. Breaks down failure root causes into tool composition and retrieval errors.

**Weaknesses:**

I appreciate the authors’ effort in creating a curated collection of MCP tools that work plug-and-play. Their main motivation is clear: existing MCP benchmarks are either too small or too difficult to use. Many tools in those benchmarks require access keys or are no longer valid. In contrast, LiveMCPBench includes key-less tools, making them easy to use. However, the benchmark itself is not large. It covers only 95 tasks, which is modest for evaluating general-purpose agentic reasoning. The number of MCP servers and tools is also limited—just 70 servers and 527 tools. This makes it unsuitable for testing agents designed for much larger MCP ecosystems with thousands of servers (something that the authors mention in the beginning of the paper).

There is another concern. Like other large benchmarks, LiveMCPBench tools may become unusable if their owners stop maintaining them. Although the toolset is packaged for reproducibility, the benchmark depends on dynamic online sources. It does not address long-term sustainability issues such as server versioning, security, or maintenance (see Limitations §A). Over time, servers may change, become unavailable, or behave differently. This can harm the reproducibility and reliability of benchmark results. The paper does not propose concrete solutions, such as Dockerized server snapshots, health checks, or semantic versioning. Without these, future evaluations may not be consistent or valid. This technical gap puts the benchmark’s long-term value for the research community at risk.

A key requirement of dealing with an 'ocean of MCP tools' is a good retrieval system that can reliably select a small superset of  tools relevant to a given task. Authors use a embedding-based retrieval subsystem which is technically rigid: it uses a single embedding model (Qwen3-Embedding-0.6B) and a single summary generator (Qwen2.5-72B-Instruct) to process server and tool descriptions, with a fixed top-k=5 retrieval (§G.2). This design choice means that all agentic models are evaluated with the same retrieval backbone, potentially conflating agent reasoning ability with retriever recall. If the retriever fails to surface relevant tools, even a strong agent may be unable to succeed. The paper’s own analysis shows that retrieval errors account for ~50% of failures (Fig. 6), but does not explore how varying the retriever, k-value, or embedding model might affect outcomes. This limits the generalizability of the findings and may overstate the impact of “retrieval errors” attributable to the chosen pipeline.

As mentioned before, LiveMCPBench includes only 95 tasks, which is modest for a benchmark intended to evaluate general-purpose agentic reasoning. Furthermore, the task distribution is skewed: 32.6% are Office tasks (Fig. 7), and annotators ultimately used only 150 out of 527 tools (28.46%; §E.1). This means that a large portion of the curated toolset remains untested, and the benchmark may over-represent office-suite workflows while under-testing other domains such as location, code, or finance. The limited and imbalanced task set restricts the breadth of agentic behaviors that can be meaningfully evaluated and may bias results toward models that excel in office-related scenarios.

The paper depends on an LLM-based automatic evaluator (LiveMCPEval) to judge task success. While this approach enables scalable, open-ended evaluation, the reported agreement with human annotators is low (see Fig. 10 and Appx. H). This means a significant fraction of judgments can diverge from human consensus, especially for long or complex trajectories. Such residual disagreement undermines the reliability of leaderboard rankings and may allow models to “game” the judge or be unfairly penalized for nuanced outputs. The paper acknowledges these limitations but does not propose robust adjudication mechanisms (e.g., dual judges, confidence calibration, or selective human audits) to mitigate the risk.

**Questions:**

1. Why did you opt for a binary success metric instead of more granular measures like retrieval recall, parameter correctness, or stability across reruns?
2. How do you ensure that the fixed retrieval pipeline (embedding model, summarizer, top-k=5) does not bias the benchmark results?
3. Have you conducted ablations with alternative retrievers, varying k-values, or hybrid retrieval methods to confirm that retrieval errors are not artifacts of your chosen pipeline?
4. Given that LiveMCPEval achieves only ~81% agreement with human judgments, what steps can you take to improve evaluator reliability?

---

> ### Author Response · Authors · 2025-11-19
>
> We thank the reviewers for their careful evaluation and constructive feedback. We have addressed all comments in detail below.
>
> 1. For **Question 1:** We use a binary task-success metric because LiveMCPBench aims to evaluate an agent’s *end-to-end* task completion ability. Fine-grained static metrics such as retrieval recall are difficult to define in our dynamic setting, where tasks can be completed through multiple valid tool-use trajectories, and they do not generalize well to extensions of the toolset or tasks.
>
>    Stability metrics like pass@k or pass^k also scale linearly with k in evaluation cost. As our benchmark includes several high-cost proprietary models, repeatedly sampling all models—especially for larger kkk—is not feasible under realistic computational constraints.
>
>    Importantly, most multi-turn agent benchmarks adopt single-run success as their primary metric. For example, SWE-Bench \[1], WebArena \[2], and TauBench \[3] (which defaults to pass@1) all rely on single-attempt evaluation.
>
>    We have added a discussion on evaluation metric choices in the revised manuscript **(Page 18, Appendix H.1, Line 938-948)**, and note that LiveMCPBench remains compatible with metrics such as pass@k or pass^k, depending on users’ computational budgets and needs.
>
> 2. For **Questions 2, 3, and Weakness 3:** To ensure that our fixed retrieval pipeline does not bias the benchmark results, we performed ablations on both the number of returned tools k and the embedding models used. These experiments aim to verify that the observed retrieval bottleneck is intrinsic to current retrieval methods rather than an artifact of our pipeline design or hyperparameter choices.
>
>    As shown in **Appendix G.3** **(Page 17, Line 885-905)**, varying k and switching embedding models yield highly consistent error patterns. This consistency suggests that retrieval-induced failures stem from fundamental limitations of existing retrieval mechanisms, not from the specific configuration we selected for the benchmark.
>
>    | Settings                    | Overall (%) |
>    |-----------------------------|-------------|
>    | Qwen3-Embedding-0.6B + k=5  | 78.95       |
>    | *Change k*                                  |
>    | Qwen3-Embedding-0.6B + k=1  | 64.21       |
>    | Qwen3-Embedding-0.6B + k=10 | 78.95       |
>    | *Change Embedding Model*                    |
>    | BGE-M3 + k=5                | 76.84       |
>
>    These ablations confirm that the retrieval bottleneck persists across parameter choices and model variants, reinforcing that the benchmark faithfully reflects current retrieval limitations rather than introducing bias.
>
> 3. For **Question 4 and Weakness 5:** Regarding the choice of the evaluation model, we have added a discussion in **Appendix H.1 (Page 18, Line 938-948)**. In this section, we examine the replaceability and robustness of our evaluation pipeline. In particular, we note that human agreement provides a practical and reliable criterion for replacing or updating the judge model using our annotated trajectory data. This human-grounded reference enables us to maintain consistency in evaluation outcomes even as the underlying judge model changes, thereby effectively mitigating concerns about evaluator stability.

---

> > ### Author Response · Authors · 2025-11-19
> >
> > 4. For **Weakness 1 and 4:** We appreciate the reviewer’s comment on the benchmark scale. LiveMCPBench was intentionally designed to be extensible. We are continuously adding newly collected tools and will periodically release expanded versions. The framework and evaluation protocol support seamless enlargement of both the toolset and tasks, requiring no additional annotation. This design ensures that the benchmark can grow alongside the rapidly evolving MCP ecosystem and remain suitable for evaluating agents at larger scales.
> >
> > 5. For **Weakness 2:** Regarding tool stability and versioning, our benchmark already enforces deterministic tool behavior by fixing specific tool versions and providing a fully reproducible Docker environment. This ensures that the execution environment remains consistent across time and across users.
> >
> >    For the stability of online MCP services, we have continuously monitored their availability and correctness since completing the toolset construction in late July 2025. To date, all MCP servers have remained functional and stable under our checks. While we acknowledge long-term sustainability as an inherent limitation of benchmarks relying on online resources, our current evidence indicates that these services are stable in practice, and our version-controlled Dockerized setup substantially mitigates concerns about reproducibility.
> >
> >
> >
> > References:
> >
> > \[1]Jimenez, C. E., Yang, J., Wettig, A., Yao, S., Pei, K., Press, O., & Narasimhan, K. R. SWE-bench: Can Language Models Resolve Real-world Github Issues?. In *The Twelfth International Conference on Learning Representations*.
> >
> > \[2]Zhou, S., Xu, F. F., Zhu, H., Zhou, X., Lo, R., Sridhar, A., ... & Neubig, G. WebArena: A Realistic Web Environment for Building Autonomous Agents. In *The Twelfth International Conference on Learning Representations*.
> >
> > \[3] Yao, S., Shinn, N., Razavi, P., & Narasimhan, K. R. (2025). $\tau $-bench: A benchmark for\underline Tool-Agent-User interaction in real-world domains. In *The Thirteenth International Conference on Learning Representations*.

---

> > ### Comment · Reviewer_d7CQ · 2025-11-20
> >
> > Some of my key concerns are still unaddressed (e.g., those in the first three paragraphs of the review), so I will maintain my score.

---

> ### Author Response · Authors · 2025-11-21
>
> Thank you for your additional feedback. We hope the following explanation will further address your concerns.
>
> 1. For the **first paragraph** (Weakness 1 in the previous response):
>
>     The reviewer raises a valid concern regarding the size of LiveMCPBench. We would like to clarify that the limited number of tasks and tools is the direct result of deliberate quality-control filtering. From an initial pool of **300 task candidates** and **5,588 MCP servers**, we retained only those that met strict criteria for reliability, reproducibility, and long-term usability—resulting in **95 high-quality tasks** and **70 robust servers**.
>
>     Beyond quantity, **diversity and functional coverage** were central to our selection process. The retained tasks span a wide range of real-world MCP usage scenarios and tool categories, which we find to be more crucial than raw scale for evaluating general-purpose agentic reasoning. As shown in Table 1, the overall benchmark size is already **comparable to several established tool-use benchmarks** (e.g., tau-bench with 165 tasks).
>
>     Finally, LiveMCPBench is explicitly designed as a **continuously extensible benchmark**. We intend to maintain and expand it over time by adding new tasks and stable MCP servers as the ecosystem grows. This ensures that, although the initial release focuses on high-quality coverage, the benchmark will progressively scale toward the large-MCP environments envisioned in the paper.
>
> 2. For the **second paragraph** (Weakness 2):
>
>     We appreciate the reviewer’s concern about long-term stability. Beyond the measures already mentioned—Dockerized packaging to ensure fixed toolset versions and ~5 months of continuous availability checks—we would like to clarify that **MCP tools are substantially more maintainable than prior API-based benchmarks** (which showed 55.6% failure within six months). MCP’s standardized schemas make failures much easier to detect and update.
>
>     Long-term maintenance is indeed unavoidable for benchmarks that rely on real tools. To address this, we plan to keep providing periodic health checks, versioned Docker snapshots for each release, and clear documentation of any tool updates. These steps ensure that past results remain reproducible and that future evaluations stay consistent.
>
>     We hope this clarifies why LiveMCPBench is both practically maintainable and more robust than previous approaches.
>
> 3. For the **third paragraph** (Weakness 3):
>
>     We hope that the newly added ablations on k and embedding models, together with the additional analysis in Appendix G.3, help address the reviewer’s concerns. **If further clarification is needed, we would greatly appreciate additional guidance** from the reviewer on how we might address this issue more thoroughly.

---

> > ### Comment · Reviewer_d7CQ · 2025-11-25
> > **Unaddressed concerns**
> >
> > 1. How do we know that a significant fraction of the MCP tools will not fail in near future? The service endpoints are maintained by 3rd parties, and they may discontinue a service or change its schema in future.
> > 2. The results are biased by the tool retrieval, as mentioned in the 3rd paragraph of my review.

---

> > > ### Author Response · Authors · 2025-11-28
> > >
> > > Thank you for the further feedback.
> > > 1. LiveMCPBench aims to measure agents’ ability to use a large set of real MCP servers in the wild. As in other benchmarks that intentionally evaluate agents on live online resources (e.g., Mind2Web 2 \[1], Online-Mind2Web \[2]), using real, large-scale MCP services is essential for ecological validity and for reflecting agents’ practical utility \[3]. Hence, **incorporating third-party online MCP servers is a deliberate and necessary choice.**
> > >
> > >    We acknowledge that some third-party services may evolve or become unavailable. To mitigate this, we rely on two safeguards: (1) a strict curation and validation pipeline, including functionality tests and schema-consistency checks; and (2) continuous health monitoring since July 2025, during which the **all of tools** have remained stable with no systematic drift observed.
> > >
> > >    Finally, LiveMCPBench is designed for long-term maintenance with versioned releases. We will refresh tool/task sets roughly every six months, while archiving each benchmark version (tasks, tools, trajectories) to ensure reproducibility and comparability over time. Deprecated endpoints will be replaced by functionally similar emerging ones to maintain the benchmark’s distribution and difficulty profile.
> > >
> > > 2. We agree that a reliable retriever is critical when operating over an “ocean of MCP tools,” and that agent performance can be bottlenecked if relevant tools are not surfaced. MCP introduces a distinctive *server–tool hierarchical structure*, which makes retrieval non-trivial: the retriever must model both query-tool relevance and intra-server tool relations. Since **MCP tool retrieval is still in an early stage** and there is no established standard, we adopt the retriever design from MCP-Zero \[4] as a representative and widely-used baseline for current MCP practice.
> > >
> > >    To address the concern on retrieval bias, we evaluate the sensitivity of our results to typical retriever choices and k values. Using **exact McNemar tests** on tasks, we find that **within a reasonable retrieval settings** (k=5 vs. k=10, or Qwen3-Embedding-0.6B vs. BGE-M3), performance differences are statistically insignificant, and **our qualitative conclusions remain unchanged**. The only exception is an intentionally extreme setting (k=1), which significantly degrades accuracy, consistent with the expectation that overly restrictive retrieval harms downstream agents.
> > >
> > >    Therefore, our findings should be interpreted as **agent performance under a representative MCP retrieval backbone (MCP-Zero)**, and they are **robust to standard variations of this backbone**. We agree that a stronger retriever could reduce the absolute failure rate and shift the error composition, and we view this benchmark as a step toward motivating and measuring such retrieval improvements in MCP. We have included this retriever sensitivity analysis in the **Appendix G.3 (Page 17)**.
> > >
> > >    | Settings                          | Overall (%) | McNemar p-value |
> > >    |-----------------------------------|-------------|-----------------|
> > >    | Qwen3-Embedding-0.6B + k=5 (main) | 78.95       | -               |
> > >    | Change k                          |             |                 |
> > >    | Qwen3-Embedding-0.6B + k=1        | 64.21       | 0.02*           |
> > >    | Qwen3-Embedding-0.6B + k=10       | 78.95       | 1.0             |
> > >    | Change Embedding Model            |             |                 |
> > >    | BGE-M3 + k=5                      | 76.84       | 0.84            |
> > >
> > >
> > > Reference:
> > >
> > > \[1] Gou, B., Huang, Z., Ning, Y., Gu, Y., Lin, M., Yu, B., … Su, Y. (2025). Mind2Web 2: Evaluating Agentic Search with Agent-as-a-Judge. *The Thirty-Ninth Annual Conference on Neural Information Processing Systems Datasets and Benchmarks Track*.
> > >
> > > \[2] Xue, T., Qi, W., Shi, T., Song, C. H., Gou, B., Song, D., … Su, Y. (2025). An Illusion of Progress? Assessing the Current State of Web Agents. *Second Conference on Language Modeling*.
> > >
> > > \[3] Anthropic team. (2025, November 24). *Introducing Advanced Tool Use on the Claude Developer Platform*.
> > >
> > > \[4] Fei, X., Zheng, X., & Feng, H. (2025). MCP-Zero: Proactive Toolchain Construction for LLM Agents from Scratch. *arXiv preprint arXiv:2506.01056*.

---

### Official Review · Reviewer_spBt · 2025-10-31

**Soundness:** 3
**Presentation:** 2
**Contribution:** 3
**Rating:** 6
**Confidence:** 4

**Summary:**

This paper introduces LiveMCPBench, a novel and challenging benchmark for evaluating LLM-powered tool-use agents within a large-scale, real-world Model Context Protocol (MCP) ecosystem.1 The authors argue that existing benchmarks are insufficient because they rely on simulated or small-scale APIs and often bypass the critical challenges of large-scale retrieval and multi-tool composition.

The key contributions include:
1. LiveMCPBench: A benchmark of 95 multi-step, dynamic "daily tasks" across six practical domains (Office, Travel, Finance, etc.).
2. LiveMCPTool: A reproducible, dependency-free tool suite comprising 70 MCP servers and 527 tools, packaged for immediate use.
3. LiveMCPEval: An automated LLM-as-a-Judge framework utilizing human-aligned key points to verify success, even with dynamic data and multiple valid solution paths.

Empirical evaluation of 10 frontier LLMs reveals a substantial performance gap (Claude-Sonnet-4 achieves 78.95% success, while most others are 30–50%). Error analysis identifies tool retrieval as the dominant bottleneck (accounting for $\sim 50\%$ of failures), and confirms a strong correlation between active tool composition and task success.

**Strengths:**

1. The paper correctly identifies and addresses the fundamental shift from small, unstable API sets to large-scale, hierarchical Model Context Protocol (MCP) ecosystems. The focus on retrieval over $500+$ tools and multi-tool composition in daily, dynamic tasks makes this benchmark uniquely challenging and representative of real-world use.
2. The curated, dependency-free toolset (LiveMCPTool) is a major engineering strength. By eliminating the reliance on scattered, proprietary API keys, the paper ensures that the benchmark is reliable and accessible, overcoming the known instability issues of prior works.
3. The empirical results are sharp and actionable. The finding that retrieval errors account for nearly 50% of failures and the positive correlation between active tool composition (measured by tool count and execution attempts) and success are clear diagnostic insights for future research on the planning and execution modules of agents.

**Weaknesses:**

1. Limited Agent Architecture Analysis: The paper uses a single baseline, the MCP Copilot Agent (a ReACT agent). While the diagnosis is excellent, the paper doesn't explore how the retrieval error issue might be mitigated by a more advanced planning or memory architecture (e.g., Tree-of-Thought, specialized memory/reflection, or workflow memory).

2. While the dependency-free nature of LiveMCPTool is a strength, the underlying data source for the 70 servers (e.g., news, weather, stock data) is only broadly mentioned as being derived from the "MCP Marketplace" and "daily scenarios." Clarifying whether these servers utilize pre-cached static data or truly access live/dynamic public APIs (even if locally hosted) would solidify the claim of dynamism.

3. The observation that the LLM-as-a-Judge effectiveness depends on the model (e.g., DeepSeek-V3 performs well, but Claude-Opus-4 performs poorly) is important. This causes a slight concern about the evaluation stability if the chosen judge model were to change or degrade. A brief discussion on judge model robustness or alternative verification protocols would strengthen this section.

**Questions:**

1. Could the authors elaborate on the implementation of the LiveMCPTool servers? Specifically, are the 70 servers accessing live, real-time public APIs (e.g., a real stock market ticker, a real weather service), or do they simulate dynamism using static but complex datasets?

2. Given that Retrieve Error is the dominant bottleneck ($\sim 50\%$), the ReACT agent used only has a single retrieval step before attempting execution. What minimal architectural modification (e.g., a Retrieval Reflection step, where the agent critiques the $k=5$ tools before acting) could be incorporated into the MCP Copilot Agent to address this weakness, and why was this approach not chosen for the baseline?

3. Cost of Composition: Table 3 shows Claude-Sonnet-4 uses $2.71$ tools and $5.59$ executions on average, demonstrating active composition. Could the authors provide a small analysis (e.g., in the Appendix) breaking down the token cost or dialogue turns associated with this active composition versus simple single-tool calls?

---

> ### Author Response · Authors · 2025-11-19
>
> We thank the reviewers for their thoughtful reviews and helpful suggestions. We provide point-by-point responses to all comments.
>
> 1. For **Question 1 and Weakness 2:** Regarding the toolset, we do not provide any static datasets nor any simulated environments. Instead, we manually verified that all tools included in the LiveMCPTool suite retrieve *genuine, real-time data* from online sources (e.g., actual live weather information or up-to-date stock prices).
>
>    To make this point explicit and to solidify our claim about dynamism, we have added a clarifying description in **Appendix F.1 (Page 16, Line 847-852)**.
>
> 2. For **Question 2 and Weakness 1:** The MCP Copilot Agent adopts a minimal and flexible architecture in which the agent autonomously performs tool retrieval and invocation. As a benchmark, our primary goal is to provide a clean and widely applicable evaluation setting, rather than to engineer additional reasoning modules into the baseline. ReACT is therefore chosen because it is the most commonly used and well-established planning framework, ensuring fair and familiar comparisons.
>
>    While adding a lightweight modification—such as a *Retrieval Reflection* step—could potentially mitigate retrieval errors, introducing such components would deviate from the principle of minimal architectural assumptions. We intentionally keep the baseline simple, so that any observed failure patterns reflect the underlying model–retrieval interaction, rather than benchmark-specific design choices.
>
>    To verify that the observed retrieval bottleneck is intrinsic to current retrieval methods—and not due to hyperparameter or embedding selection—we conducted ablations on returned tools (k) and embedding models. Results **(Page 17, Appendix G.3, Line 885-905)** show consistent error trends across settings, indicating that the limitations are a property of existing retrieval mechanisms and not a consequence of our benchmark design.
>
>    | Settings                    | Overall (%) |
>    |-----------------------------|-------------|
>    | Qwen3-Embedding-0.6B + k=5  | 78.95       |
>    | *Change k*                                  |
>    | Qwen3-Embedding-0.6B + k=1  | 64.21       |
>    | Qwen3-Embedding-0.6B + k=10 | 78.95       |
>    | *Change Embedding Model*                    |
>    | BGE-M3 + k=5                | 76.84       |
>
>
>
> 3. For **Question 3:** We have updated Table 3 to include the token consumption of each model. In addition, we added a new analysis in **Appendix G.4 (Page 17, Line 908-917)**, where we break down the average number of steps, tool invocations, and token usage. Our findings indicate that, beyond the number of steps, active tool exploration is also a significant contributor to increased token cost.
>
> 4. For **Weakness 3:** Regarding the choice of the evaluation model, we have added a dedicated discussion in **Appendix H.1 (Page 18, Line 938-948)**. In this section, we examine the applicability of alternative metrics (e.g., pass@k and pass^k) within our framework and analyze how they can complement or substitute LLM-based judgments. Furthermore, we discuss the replaceability and robustness of the evaluation system. Specifically, we highlight that human agreement can serve as a practical criterion when replacing or updating the judge model on our annotated trajectory data. This provides a stable reference for maintaining evaluation consistency as the underlying judge model evolves, thereby mitigating concerns about evaluation stability.

---

### Official Review · Reviewer_i9Xk · 2025-10-31

**Soundness:** 2
**Presentation:** 2
**Contribution:** 3
**Rating:** 4
**Confidence:** 3

**Summary:**

This paper introduces LiveMCPBench, a benchmark for evaluating LLM agents in large-scale MCP environments. It comprises (1) 95 real-world tasks, (2) LiveMCPTool with 70 servers and 527 tools, (3) LiveMCPEval achieving 81% human agreement for automated evaluation, and (4) MCP Copilot Agent as baseline. Testing 10 frontier models reveals Claude-Sonnet-4 reaches 78.95% success rate, while most models struggle with retrieval errors (50%) and meta-tool-learning in multi-tool scenarios.

**Strengths:**

1. Realism and scale: LiveMCPBench captures the complexity of real MCP ecosystems with 527 tools across 70 servers, moving beyond prior single-server or simulated tool-use settings.
2. Well-rounded design: The integration of LiveMCPTool, LiveMCPEval, and MCP Copilot Agent provides a coherent, end-to-end evaluation framework that is both scalable and reproducible.
3. Empirical insight: The benchmark yields meaningful diagnostic findings—retrieval errors dominate failure modes, and tool composition correlates strongly with success.
4. Promised reproducibility: The work offers a ready-to-deploy tool suite and automated evaluation protocol, enabling consistent, and the authors intend to release code and data.

**Weaknesses:**

1. Limited model coverage: Only ten models are tested, excluding several widely used or newly released systems (e.g., GPT-5, Grok-4, GLM-4.5). This undermines the generalizability of the reported findings.
2. Volatile metric design: The experiment employs a single-pass success metric. Realistic deployments often depend on reliability across multiple attempts. Using metrics such as pass@k (success in k runs) and pass^k (all k runs successful) would better capture stability and practical robustness.
3. Evaluation methodology limitations: While LiveMCPEval enables scalable automatic assessment via the LLM-as-a-Judge paradigm, up to a 10-point discrepancy from human annotations indicates that purely LLM-based evaluation may be insufficient for structured MCP interactions. Integrating rule-based or code-verification modules might improve transparency, interpretability, and consistency with MCP's deterministic design.
4. Missing annotation and cost detailed demonstration: As a benchmark paper, it lacks concrete examples demonstrating the annotation process and a detailed report of API token usage and computational cost, which weakens transparency and reproducibility.
5. Typo in paper writing: In Figure 11, "Evaulation Error" should be corrected to "Evaluation Error."

**Questions:**

1. How were the ten evaluated models selected, and are there plans to include more recent systems such as GPT-5, Grok-4, or GLM-4.5 to improve completeness and ensure broader coverage across model families?
2. Why did the evaluation rely solely on single-run success rates instead of including reliability metrics such as pass@k or pass^k that better capture model stability under repeated execution? Could such metrics be incorporated in future updates of the benchmark?
3. Can you provide a concrete annotation example illustrating how the annotation principles are applied in practice, along with a detailed summary of API token usage of LiveMCPBench?
4. What criteria were used to define task difficulty across the 95 daily tasks, and how well do these categories align with real-world MCP usage complexity?
5. Could the benchmark be extended to support multi-agent collaboration scenarios, where multiple LLMs coordinate across MCP servers? If so, what challenges or modifications would be required in the current setup?

---

> ### Author Response · Authors · 2025-11-19
>
> We sincerely appreciate the reviewers’ careful reading of our manuscript and their constructive feedback. We have carefully addressed the comments and corrected all typos in the revised version.
>
> 1. For **Question 1 and Weakness 1:** We selected the ten evaluated models based on the **state-of-the-art systems available by the end of July 2025**, ensuring that our comparisons were conducted among models that were regarded as competitive at that time. To maintain broad coverage across different model families, we continuously benchmark newly released models as they become accessible.
>
>    Since completing the initial submission, we have already evaluated several more recent systems, including **Grok-4, GLM-4.5, and GPT-5**. We will continually update the benchmark to reflect new model releases, ensuring our evaluation remains comprehensive and current.
>
>    Below, we provide the results for the newly added models for reference:
>
>     | Model             | Office | Leisure | Travel | Lifestyle | Finance  | Shopping  | Overall (%) |
>     | ----------------- | ------ | ------- | ------ | --------- | -------- | --------- | ----------- |
>     | Claude-Sonnet-4   | 90.32  | 64.29   | 75.00  | 80.0 0    | 78.57    | 66.67     | 78.95       |
>     | Grok-4 **(new)**  | 77.42  | 50.00   | 75.00  | 60.00     | 78.57    | 44.44     | 67.37       |
>     | GLM-4.5 **(new)** | 80.65  | 42.86   | 41.67  | 60.00     | 64.29    | 44.44     | 61.05       |
>     | GPT-5 **(new)**   | 54.84  | 50.00   | 41.67  | 53.33     | 64.29    | 44.44     | 52.63       |
>     | DeepSeek-R1       | 41.94  | 50.00   | 58.33  | 46.67     | 50.00    | 55.56     | 48.42       |
>
> * For **Question 2 and Weakness 2:** The choice of evaluation metrics is closely tied to the intended usage scenario of the benchmark. We adopted single-run task success rates because they provide the most straightforward and accessible means of comparing agent performance, and they are widely used as the primary metric in existing multi-step agent evaluation frameworks. Metrics such as *pass@k* offer a better characterization of an agent’s discovery ability across multiple attempts, while *pass^k* focuses on stability and consistency under repeated executions. Both metrics indeed provide a more comprehensive view of agent reliability.
>
>   Methodologically, these metrics can be incorporated into our benchmark without requiring any modification to task definitions or the evaluation protocol. However, their computational cost grows linearly with *k*. Since our evaluation includes several proprietary large models that incur substantial API costs, performing repeated rollouts for all frontier models—especially at larger *k* values—was infeasible under our resource constraints.
>
>   It is worth noting that most complex multi-turn agent benchmarks also rely primarily on single-run success rates for comparison. For example, SWE-Bench \[1] and WebArena \[2] adopt single-attempt success as their main metric, and TauBench \[3] similarly uses pass@1 as the default comparison metric:
>
>   > *"By default, we report the average reward across tasks, pass^1 = pass@1 = E\[r] = E\[c/n], as the main metric for comparing agents."*
>
>   We have added a discussion on evaluation model selection and metric design in the revised manuscript **(Page 18, Appendix H.1, Line 938-948)**, and we encourage users to choose appropriate metrics—such as *pass@k* or *pass^k*—depending on their computational budget and application needs.
>
> * For **Question 3 and Weakness 4:** We have added an example diagram of the annotation process in **Fig. 8 (Page 15)** to show how the annotation principles are applied. We also updated **Table 3 (Page 7)** to include the detailed token usage for each model.

---

> > ### Author Response · Authors · 2025-11-19
> >
> > * For **Question 4:** For an agent, task difficulty can be approximated using two measurable factors: the *average number of key points* annotated by humans and the *average number of tools* required. In LiveMCPBench, these averages are 2.82 and 2.69, respectively, reflecting the multi-step nature of daily tasks.
> >
> >   In LiveMCPBench, we instead categorize tasks into six representative real-world scenarios and employ verifiers to ensure annotation quality. We also confirm that tasks are solvable with our curated toolset, collected and filtered from real-world MCP servers, ensuring good alignment with actual MCP usage complexity.
> >
> > * For **Question 5:** Our task design and toolset can be naturally extended to multi-agent settings, where multiple LLMs collaborate across MCP servers. However, fully *automating* the evaluation of such multi-agent interactions remains an open challenge. In particular, how to reliably assess performance based on multi-agent behavioral trajectories requires further investigation and methodological development.
> >
> > * For **Weakness 3:** In our dynamic setting, tasks can be completed through multiple valid trajectories, which makes it difficult to define consistent rule-based evaluations that remain stable. We have updated **Appendix H.1 (Page 18, Line 938-948)** to include additional discussion on how principled evaluation criteria—beyond purely model-based scoring—can be incorporated and how different classes of judge models can be selected or combined.
> >
> > References:
> >
> > \[1]Jimenez, C. E., Yang, J., Wettig, A., Yao, S., Pei, K., Press, O., & Narasimhan, K. R. SWE-bench: Can Language Models Resolve Real-world Github Issues?. In *The Twelfth International Conference on Learning Representations*.
> >
> > \[2]Zhou, S., Xu, F. F., Zhu, H., Zhou, X., Lo, R., Sridhar, A., ... & Neubig, G. WebArena: A Realistic Web Environment for Building Autonomous Agents. In *The Twelfth International Conference on Learning Representations*.
> >
> > \[3] Yao, S., Shinn, N., Razavi, P., & Narasimhan, K. R. (2025). $\tau $-bench: A benchmark for Tool-Agent-User interaction in real-world domains. In *The Thirteenth International Conference on Learning Representations*.

---

### Comment · Reviewer_kNq6 · 2025-11-28

Hi AC, I would like to adjust my score for this submission, but it seems that the system no longer allows me to modify the rating. Could you please check if this is a system restriction or if scoring has already been locked? Thank you!

---

> ### Comment · Area_Chair_CVFR · 2025-11-28
>
> Hi Reviewer,
>
> I am not 100% sure. It may or may not be because a significant bug that Openreview is fixing recently. Please try it later.
>
> Best,
> AC

---

### Author Response · Authors · 2025-11-30
**General Response**

1. Response on newer systems (Reviewer i9Xk)

   We evaluated several more recent model systems. The results are shown below.

   | Model             | Office | Leisure | Travel | Lifestyle | Finance  | Shopping  | Overall (%) |
   | ----------------- | ------ | ------- | ------ | --------- | -------- | --------- | ----------- |
   | Claude-Sonnet-4   | 90.32  | 64.29   | 75.00  | 80.0 0    | 78.57    | 66.67     | 78.95       |
   | Grok-4 **(new)**  | 77.42  | 50.00   | 75.00  | 60.00     | 78.57    | 44.44     | 67.37       |
   | GLM-4.5 **(new)** | 80.65  | 42.86   | 41.67  | 60.00     | 64.29    | 44.44     | 61.05       |
   | GPT-5 **(new)**   | 54.84  | 50.00   | 41.67  | 53.33     | 64.29    | 44.44     | 52.63       |
   | DeepSeek-R1       | 41.94  | 50.00   | 58.33  | 46.67     | 50.00    | 55.56     | 48.42       |

2. Response on retrieval-system ablations (Reviewers spBt, d7CQ)

   We ablated (1) the number of returned tools k and (2) the embedding model in the retrieval system. We conducted exact McNemar tests and found that our default setting does **not** significantly affect the conclusions.

   | Settings                          | Overall (%) | McNemar p-value |
   |-----------------------------------|-------------|-----------------|
   | Qwen3-Embedding-0.6B + k=5 (main) | 78.95       | -               |
   | Change k                          |             |                 |
   | Qwen3-Embedding-0.6B + k=1        | 64.21       | 0.02*           |
   | Qwen3-Embedding-0.6B + k=10       | 78.95       | 1.0             |
   | Change Embedding Model            |             |                 |
   | BGE-M3 + k=5                      | 76.84       | 0.84            |


3. Response on evaluator reliability (Reviewers i9Xk, spBt, kNq6)

   In addition to the human agreement already reported (using DeepSeek-V3, \~80%) and the quantitative error analysis in Appendix H.2 (Page 18), we added two new pieces of evidence during rebuttal.

   **Evaluator bias:** When using a suitable evaluator model, **the relative ranking of systems is stable**. We replaced DeepSeek-V3 with Qwen-2.5-72B-Instruct and tested ranking stability using Kendall’s τ-b: τ-b = 0.8864, p = 0.0004 (< 0.001), showing statistically significant consistency.

   **Tool hallucination under trajectory-based evaluation:&#x20;**&#x54;he overall tool-hallucination rate is 9.00%, which is relatively low. The miss-detection rate for tool hallucination is 27.3%, accounting for only 1.6% of all tasks. Thus, even without a post-validation layer, **the current protocol already catches the majority of such failures**, and the remaining undetected cases form a very small upper bound on evaluation noise.

1. Response on metric choice (Reviewers i9Xk, d7CQ)

   We clarify that pass@k and pass^k can be directly integrated into our current evaluation pipeline. However, because some proprietary models are very expensive, we are unable to compute these metrics for large k, which limits their practical use in our setting.

   To address this, we further evaluated pass@maj and analyzed when it is appropriate for our method. Specifically, we tested multi-evaluator ensembles on annotated Claude-Sonnet-4 trajectories:

   * **pass@maj** achieves 77.89% human agreement (better than the weakest evaluator DeepSeek-R1 at 58.95%, slightly below DeepSeek-V3 at 81.05%).

   * **At least one evaluator correct** reaches 97.89%.

   These results provide preliminary evidence that **multi-evaluator voting can be reliable even without prior knowledge of evaluator quality**.

---

### Meta-Review · Area_Chair_deca · 2026-01-07

**Summary:**

This paper's contribution is the LiveMCPBench to evaluate LLM agent tool-use capability in MCP environment. It has 95 tasks, 70 MCP servers and 527 tools. The insight is that retrieval is a major bottleneck for success. The contribution is very good overall.

This paper received scores of 4,4,6,8. There are some concerns that are not well addressed by the authors such as the unstableness of the benchmark and the lacking of diverse scenarios and limited tools that are actually used (see comments in the box below).

The AC thinks this benchmark paper is not at the level of ICLR and would recommend reject.

**Reviewer Concerns:**

One of the major concerns is that "LiveMCPBench tools may become unusable if their owners stop maintaining them". While authors made a statement that the tools remain stable in a 6-month window, it is still unconvincing. The same reviewer also mentioned that a large portion of the 500+ tools are not being used and that there is a bias towards office scenario. This might not be a good improvement over tau-bench.

**Reviewer Scores:**

The last reviewer's final decision is 8, while other reviewers didn't have the chance to update their scores.
The AC thinks that the two negative reviewers are unlikely to improve their scores - one reviewer explicitly mentioned this during discussion. The reviewer who gave 6 also mentioned that experiments should use other agent frameworks than ReAct. So it's unlikely they will improve the score over 6.

So the AC thinks that the paper's final scores would've been mixed.

---

### Decision · Program_Chairs · 2026-01-26

Reject